# Nitrogen Cycling in CMIP6 Land Surface Models: Progress and Limitations

Taraka Davies-Barnard[1,2], Johannes Meyerholt[2], Sönke Zaehle[2], Pierre Friedlingstein[1,3], Victor Brovkin[4], Yuanchao Fan[5,6], Rosie A. Fisher[7,8], Chris D. Jones[9], Hanna Lee[5], Daniele Peano[10], Benjamin Smith[11,12], David Wårlind[11], and Andy Wiltshire[9]

[1]University of Exeter, Exeter, UK
[2]Max Planck Institute for Biogeochemistry, Jena, Germany
[3]Laboratoire de Meteorologie Dynamique, Institut Pierre-Simon Laplace, CNRS-ENS-UPMC-X, Departement de Geosciences, Ecole Normale Superieure, 24 rue Lhomond, 75005 Paris, France
[4]Max Planck Institute for Meteorology, Hamburg, Germany
[5]NORCE Norwegian Research Centre, Bjerknes Centre for Climate Research, Bergen, Norway
[6]Harvard University, Cambridge, USA
[7]National Center for Atmospheric Research, Boulder, Colorado, USA
[8]Centre Européen de Recherche et de Formation Avancée en Calcul Scientifique, Toulouse, France
[9]Met Office Hadley Centre, Exeter, UK
[10]Fondazione Centro euro-Mediterraneo sui Cambiamenti Climatici, Bologna, Italy
[11]Department of Physical Geography and Ecosystem Science, Lund University, Lund, Sweden
[12]Hawkesbury Institute for the Environment, Western Sydney University, Richmond, Australia

*Correspondence to*: T. Davies-Barnard (t.davies-barnard@exeter.ac.uk)

**Abstract.** The nitrogen cycle and its effect on carbon uptake in the terrestrial biosphere is a recent progression in earth system models. As with any new component of a model, it is important to understand the behaviour, strengths, and limitations of the various process representations. Here we assess and compare five land surface models with nitrogen cycles that are used as the terrestrial components of some of the earth system models in CMIP6. The land surface models were run offline with a common spin-up and forcing protocol. We use a historical control simulation and two perturbations to assess the models' nitrogen-related performance: a simulation with atmospheric carbon dioxide increased by 200 ppm, and one with nitrogen deposition increased by 50 kgN ha$^{-1}$ yr$^{-1}$. There is generally greater variability in productivity response between models to increased nitrogen than to carbon dioxide. Across the five models the response to carbon dioxide globally was 5 to 20% and the response to nitrogen was 2 to 24%. The models are not evenly distributed within the ensemble range, with two of the models having low productivity response to nitrogen, and another one low response to elevated atmospheric carbon dioxide, compared to the other models. In all five models individual grid cells tend to exhibit bimodality, with either a strong response to increased nitrogen or atmospheric carbon dioxide, but rarely to both to an equal extent. However, this local effect does not scale to either the regional or global level. The global and tropical responses are generally more accurately modelled than boreal, tundra, or other high latitude areas compared to observations. These results are due to divergent choices in the representation of key nitrogen cycle processes. They show the need for more observational studies to enhance understanding of nitrogen cycle processes, especially nitrogen-use efficiency and biological nitrogen fixation.

## 1 Introduction

The terrestrial carbon (C) cycle currently removes around a third of anthropogenic carbon emissions from the atmosphere (Friedlingstein et al., 2019; Le Quéré et al., 2018). Changes in this uptake will affect the allowable emissions (Seneviratne et al., 2016) for targets such as limiting warming to 1.5°C (Millar et al., 2017; Müller et al., 2016). Nitrogen (N) is required to synthesise new plant tissue (biomass) out of plant-assimilated C, in differing ratios across biomes and tissue types (McGroddy et al., 2004). Therefore, future projections of terrestrial C uptake are dependent on N availability, particularly under high atmospheric carbon dioxide ($CO_2$) conditions (Arora et al., 2020; Meyerholt et al., 2020; Wieder et al., 2015b; Zaehle et al., 2014b). A key tool for projections of allowable emissions are Earth System Models (ESMs), which project the responses of the coupled earth system to perturbations in forcings (Anav et al., 2013; Arora et al., 2013; Friedlingstein et al., 2006; Jones et al., 2013). Of the ESMs that contributed results to the Fifth Phase of the Coupled Model Intercomparison Project (CMIP5, Taylor et al., 2012) only two, based on the same land component, included terrestrial N cycling (Thornton et al., 2009). A number of studies with stand-alone terrestrial biosphere models (Sokolov et al., 2008; Wårlind et al., 2014; Zaehle et al., 2010; Zhang et al., 2013) as well as post-hoc assessments of CMIP5 projections suggest that predictions of terrestrial C uptake would decrease by 37 – 58% if ESMs accounted for N constraints (Wieder et al., 2015b; Zaehle et al., 2014b).

Among the latest generation of models contributing results to CMIP6 (Eyring et al., 2016) at least ten ESMs incorporate the N cycle (Arora et al., 2020). These models employ a range of assumptions and process formulations, reflecting divergent theory and significant knowledge gaps (Zaehle and Dalmonech, 2011). Initial results imply that the inclusion of an N cycle has reduced the spread of results across multiple ESMs (Jones and Friedlingstein, 2020). Since N availability is an important source of uncertainty for the C cycle, (Meyerholt et al., 2020) an assessment of the sensitivity of the N cycle in these models to changes in atmospheric $CO_2$ and N inputs is required. Because of the tight coupling of C and N dynamics, a direct evaluation of the N effects on simulated C cycle dynamics using conventional model benchmarking approaches (Collier et al., 2018; Luo et al., 2012) is challenging. More insights into the magnitude of a N effect can be gained by comparing model simulations against perturbation experiments that provide evidence for the responses of terrestrial ecosystems to changes in the C and N availability (Thomas et al., 2013; Wieder et al., 2019; Zaehle et al., 2010).

In this study, we test five land surface models (LSMs) employed in the latest generation of ESMs used in CMIP6. We use a set of standardised model forcing and protocol to simulate historical changes in the C and N balance, as well as the response to N and C perturbations. The perturbation experiments (described in the methods) are designed to approximate field experiments undertaken to understand the effects of elevated $CO_2$ or N (e.g. Ainsworth and Long, 2005; LeBauer and Treseder, 2008; Song et al., 2019). These simulations reveal the overall pattern of response of the model to these forcings. We use a range of observations from the literature and model-to-model comparisons to assess the behaviour and

performance of the models. Comparisons between models alone can also provide useful insight into the models' behaviour.

The approach of assessing ESM N cycles via their corresponding offline LSMs, driven by a standardised set of model forcing, has the advantage of making model projections directly comparable while giving a representative view of the latest N cycle developments.

## 2 Methods

### 2.1 Models

We ran simulations with five LSMs that are the land components of ESMs taking part in CMIP6. The key N process formulations are summarized in Table 1. A brief description of each model follows.

The Community Land Model version 4.5 (CLM4.5; Koven et al., 2013; Oleson et al., 2010) is used in the Euro-Mediterranean Centre on Climate Change coupled climate model (CMCC-CM2; Cherchi et al., 2019) and TaiESM1. The N component is described in Koven et al., (2013). CLM4 is the precursor to CLM4.5 and was the first N model for ESMs, used

in CMIP5 (Thornton et al., 2007, 2009). While the N cycling component of CLM4.5 is similar to CLM4, some features of CLM4.5, such as leaf physiological traits (Bonan et al., 2012), were modified and there is a vertically resolved soil biogeochemistry scheme (Koven et al., 2013) as opposed to the single-layer box modelling scheme for C and N in CLM4.

The Community Land Model version 5 (CLM5; Lawrence et al., 2019) is used in The Community Earth System Model Version 2 (CESM2; Danabasoglu et al., 2020) and the Norwegian Earth System Model version 2 (NorESM2; Seland et al.,

2020). CLM5 is the latest version of CLM and represents a suite of developments on top of CLM4.5. The N component is described in Fisher et al., (2010); and Shi et al., (2016). The key difference for the N cycle compared to CLM4 is the implementation of a C cost basis for acquiring N, derived from the Fixation and Uptake of Nitrogen (FUN) approach (Fisher et al., 2010).

JSBACH version 3.20 model (Goll et al., 2017) is used in the Max Planck Earth System Model version 1.2 (MPI-ESM;

Mauritsen et al., 2019) and Alfred Wegener Institute Earth System Model (AWI-ESM). The N component is described in Goll et al., (2017).

The Joint UK Land Environment Simulator version 5.4 (JULES-ES; Best et al., 2011; Clark et al., 2011) is used in the UK Earth System Model (UKESM1; Sellar et al., 2020.). The N component is described in Wiltshire et al., (2020) and Sellar et al., (2020).

The Lund-Potsdam-Jena General Ecosystem Simulator version 4.0 (LPJ-GUESS; Olin et al., 2015; Smith et al., 2014) is used in the European community Earth-System Model (EC-Earth; Hazeleger et al., 2012). The N component is described in Smith et al., (2014).

## 2.2 Forcing Data and Model Initialisation

All models' pools were spun-up to equilibrium forced by pre-industrial conditions. This comprised of a constant atmospheric $CO_2$ concentration of 287.14 ppm, cycling global climate data at 0.5° x 0.5° resolution for the years 1901-1930 from the CRU-NCEP dataset version 7.0 (New et al., 2000), using constant 1860 land cover from the Hurtt et al., (2020) database, and 1860s nitrogen deposition from the Atmospheric Chemistry and Climate Model Intercomparison Project (Lamarque et al., 2013). Next, transient historical runs were performed for the 1861-1900 period with the same climate forcing as the spin-up, but with time-varying atmospheric $CO_2$ concentrations from synthesized ice core and National Oceanic and Atmospheric Administration (NOAA) measurements, as well as annually varying land-use from Hurtt et al., (2020). The N deposition is taken from the Atmospheric Chemistry and Climate Model Intercomparison Project (Lamarque et al., 2013). The simulations were then continued for 1901 – 2015 under all time-varying forcings, including climate.

The models applied their individual soil and vegetation spin-ups according to their respective conventions. The goal of the spin-up procedure is to obtain quasi-steady states of the ecosystem pools in relation to climate, avoiding drifting pool sizes due to lack of equilibrium, especially for slow-turnover soil organic matter pools. Because of differences among the models, pool sizes after spin-up are not expected to be identical.

## 2.3 Model Experiments

In addition to the historical run described above (referred to hereafter as the Control), two experiments were performed for the period 1996-2015: increased $CO_2$ (+CO2) and increased N (+N). These two experimental runs are compared to the corresponding 1996-2015 simulations from the unperturbed Control runs. SI Table 1 provides a summary of the experiments. For the increased $CO_2$ experiment (+CO2) the atmospheric $CO_2$ concentration was abruptly increased to constant 550 ppm. This is almost twice the pre-industrial atmospheric $CO_2$ of 280 ppm or a 200 ppm increase compared to the 1996 atmospheric $CO_2$ of ~350 ppm, similar to free-air $CO_2$ enrichment experiments performed in the 1990s (Norby et al., 2005). For the increased N experiment (+N) N deposition was abruptly increased by 50 kgN ha$^{-1}$ yr$^{-1}$, which is roughly equivalent to what has been used in a number of forest N fertilisation trials (Thomas et al., 2013) and around 5 – 10 times higher than typical background N deposition (Zak et al., 2017).

## 2.4 Analytical Framework

The response of the terrestrial productivity (and with it terrestrial C storage) to changes in the N cycle is in principle controlled by two components: (i) the net ecosystem balance of N, i.e. the difference between changes in ecosystem N inputs and N losses, which determines the change in the ecosystem N available for plant growth and immobilisation during litter and soil organic matter decomposition, and (ii) the ratio of carbon production per unit N availability, which can be most effectively be described as the N-use efficiency of growth.

Because the individual processes and pools considered varies between the five models (Table 1), we use a simplified N budget to assess the annual change in the terrestrial N store ($\Delta N$, including soil and plants):

$$\Delta N = N_{dep} + BNF - N_{loss} \qquad\qquad\qquad (1)$$

where $N_{dep}$ is the N deposition, BNF is the biological N fixation, and $N_{loss}$ is the N lost from gaseous, leaching, and other pathways, as declared by the models. This paradigm assumes that increased ecosystem N input from deposition or fixation enters the soil and then becomes available for plant uptake. In a similar way, plant N uptake ($N_{up}$) could lead to reduced N losses, which would (assuming constant N inputs) result in an apparent increase in the ecosystem N capital. Note that crop fertilisation is not included here, as it is assumed to be equal in the 3 simulations.

Whether and how this change in N capital affects plant growth is dependent on the magnitude of the change in plant N uptake, as well as relationship between $N_{up}$ and NPP (whole-plant nitrogen-use efficiency; NUE; (Zaehle et al., 2014a))

$$NUE = \frac{NPP}{N_{up}} \qquad . \qquad\qquad\qquad (2)$$

where $N_{up}$ includes plant uptake of soil inorganic N of any origin, i.e. atmospheric deposition, fertilization, decomposition of plant litter, or biological nitrogen fixation (BNF). NUE is the outcome of the product of tissue stoichiometry and fractional allocation of NPP to different tissue types, and therefore varies with changes in the allocation fractions and tissue C:N.

### 2.5 Observations for Comparison

We compare the models to a range of observation-based metrics at global and regional scales, detailed in SI Table 2. Most of the numbers from the literature that we cite are based on relatively small numbers of field studies upscaled or averaged to give an approximate global value with confidence intervals. No modification of spatial scale or averaging is done to values used, but where the $CO_2$ or N increase is specified it is scaled to 200 ppm or 50 kg ha$^{-1}$ yr$^{-1}$ accordingly. While these upscaled values need to be interpreted with caution, in the absence of more robust comparators they are useful benchmarks that can provide real-world context in addition to field scale comparisons and inter-model comparisons. Where appropriate, comparisons are made at the climate-determined region level (see SI Fig. 1; (Kottek et al., 2006)).

### 3 Results

### 3.1 Control Run Global C and N budgets

A range of pools and fluxes from the models compared to the closest comparable observation-based data show a good performance overall and emphasises similarities between the models at the global scale (Fig. 1). For GPP, all the models

compare well to the MTE data (Jung et al., 2011) and when the directly comparable time period is used (see SI Fig. 2) the models are all within the MTE range. The global GPP value is underlain by some regional variations between models (SI Fig. 2 and 3).

Like GPP, the total ecosystem respiration (TER) is similar across all the models and most of the models fall within the range of a top-down estimate by Ballantyne et al., (2017) ($106 \pm 12$ GtC yr$^{-1}$). However, the partitioning between the autotrophic and heterotrophic respiration differs (Fig. 1). Autotrophic respiration is overestimated in all the models (Luyssaert et al., 2007; Piao et al., 2010), while heterotrophic respiration is underestimated (Bond-Lamberty and Thomson, 2010). The heterotrophic value from Bond-Lamberty and Thomson, (2010) was reduced by 33% to account for root respiration in line with Bowden et al., (1993).

N inputs differ strongly between the models because of widely varying biological nitrogen fixation (BNF, Fig. 1). The other major input, N deposition, is a prescribed input with small variations resulting from differences in the land-sea mask of the individual models. BNF on the other hand has a wide range among models. An upscaled meta-analysis of BNF covering the period of approximately 1990 – 2019 (Davies-Barnard and Friedlingstein, 2020) has a range of 52 – 130 TgN yr$^{-1}$ and only one model is outside of that range. The three models with the highest BNF (JSBACH, CLM5, and JULES-ES) are three of the four models that use an NPP based function (the fourth being CLM4.5). CLM5's process-based function uses a C cost of N acquisition where energy from NPP can produce N based on the work by Fisher et al., (2010). JULES-ES, JSBACH, and CLM4.5 use an empirical large-scale correlation with NPP (Cleveland et al., 1999). LPJ-GUESS, the lowest BNF model, also uses an empirical correlation from Cleveland et al., (1999), based on evapotranspiration rather than NPP. Thus, even BNF functions based on the same source (Cleveland et al., 1999) can have very different results (Wieder et al., 2015a), due to the large range of BNF functions within the source and differences in how they are implemented (Meyerholt et al., 2016). BNF dominates N input variability both because of lack of process understanding to constrain model structures and the continued uncertainty in available observations.

Looking at the soil and vegetation C and N pools and the ratios between them, the models have a range of strengths and weaknesses, with no model falling within the observation-constrained range for all pools. However, due to variations in both the modelling and measurement of C and N within different soil depths, not too much emphasis should be placed on the pool comparisons shown in Fig. 1.

**3.2 Modelled NPP Responses to +CO2 Experiment**

The ensembles' global modelled response of NPP to +CO2 concurs with a meta-analysis of NPP responses to +200 ppm $CO_2$ suggests a positive response of $15.6 \pm 12.8\%$ (Song et al., 2019) (Table 2), with all models within that range. Other meta-analyses of productivity (for instance, aboveground woody biomass) changes associated with elevated $CO_2$ give higher ranges of response (Table 2). These other measures of productivity suggest a lower limit of around 12%, which encompasses all but one of the models. However, models falling within the range of the observations may be equally indicative of biases

and lack of precision in the observational estimates, as the fidelity with which the models can predict local and global response to elevated $CO_2$.

CLM4.5 has a notably lower NPP response to +CO2 than the other models (Fig. 2), with the exception of areas where the absolute magnitude of NPP is very low and small absolute changes (SI Fig. 4) already lead to large proportional changes. However, even in these regions, the absolute changes are consistently less than the other four models (SI Fig. 4). The low

response in CLM4.5 is due to a lack of mechanisms to ameliorate N limitation when C supply increases, for instance via variable C:N ratios or increased BNF (as is the case for CLM5) (Fisher et al., 2018; Wieder et al., 2019). This strong limitation by the N cycle was a key reason why CESM and NorESM in CMIP5 had lower C uptake in response to $CO_2$ compared to other carbon cycle ESMs (Arora et al., 2013).

Despite the seeming agreement of the NPP response to +CO2 at the global scale, the regional patterns in response vary

considerably for key biomes (Fig. 2). In high latitude tundra areas, the +CO2 response ranges between near zero (JULES-ES), very low in CLM4.5, JSBACH and LPJ-Guess to high (CLM5). In most models, this region shows sparse vegetation cover and nitrogen availability, allowing for only little increase in response to elevated $CO_2$, whereas the increased BNF in CLM5 facilitates a response to increasing $CO_2$ levels. With the exception of JULES-ES, most models predict a large +CO2 response in very dry ecosystems with marginal productivity.

The NPP response of the equatorial region overall (SI Table 3 and SI Fig. 1) to +CO2 ranges from 5% for CLM4.5 to 23% for CLM5 and JSBACH. Looking at latitudinal averages (SI Fig. 4) we can see the overall patterns are consistent across most models, and while the percent change varies a lot, the absolute change in NPP shows considerable agreement between models, with the exception of CLM4.5. Model responses of NPP to +CO2 in greater Amazonia however, do not reach a consensus. Comparing the response in the Amazonia region with that of coastal regions of northern South America, the

JSBACH response is lower, CLM5 and LPJ-GUESS higher, and JULES-ES and CLM4.5 are approximately the same. JSBACH's dip in +CO2 NPP response at the equator (compared to surrounding areas) can also be seen in the absolute values averaged by latitude (SI Fig. 4). The process responsible for this spatial pattern is currently unclear, but may be associated with the strongly enhanced GPP simulated by the model for this region compared to observation-derived estimates (SI Fig. 2).


## 3.3 Modelled NPP Responses to +N Experiment

The response to +N in the models shows a binary distribution, with models exhibiting either a high (>20%) or low (<3%) response (Fig. 3) at the global scale. A meta-analysis of NPP responses to +50 kg N ha$^{-1}$ yr$^{-1}$ suggests a positive response of 3 – 10.5% (Song et al., 2019) but none of the models are within this range (Table 2.). Other meta-analyses of productivity

changes with increased N give higher ranges of response (11 – 39.8%), encompassing three of the five models (Table 2). As both a percent change and absolute change (see SI Fig. 5) JULES and JSBACH show much lower +N NPP response than the

other models considered here. CLM4.5 has the highest response (24%), on account of its high initial N limitation (Koven et al., 2013).

The tundra biome response is high in CLM5 and JULES-ES, and lower but present in LPJ-GUESS and CLM4.5 (Fig. 3 and
SI Fig. 5). If low NPP is excluded then the tundra mean response across models is 2 – 9% (SI Table 3) much lower than the average of observations compiled by LeBauer and Treseder, (2008) of 35% (95% confidence interval 12 – 64%). There is a high response to +N in Africa & Australia in CLM4.5, CLM5, and LPJ-GUESS, despite aridity likely limiting increase in NPP in absolute, if not relative, terms, but insufficient observations to make meaningful comparisons. One area of agreement between the models is the lack of +N response of the Amazonian region (Fig. 3) which is consistent with observations which
show just a 5% +N response in tropical forests (Schulte-Uebbing and Vries, 2018). However, when other tropical regions are included in the models the +N NPP response rises to 17 – 20% in LPJ-GUESS, CLM4.5 and CLM5, with JULES-ES and JSBACH remaining low (SI Table 3).

### 3.4 Comparison of NPP +N and +CO2 Responses

It might be anticipated that there would be a relationship between the +N and +CO2 responses, as an ecosystem (model) that is less N limited could respond more strongly to increased atmospheric $CO_2$ (Meyerholt et al., 2020). A lack of response to N fertilisation could indicate sufficient N supply and therefore a lacking constraint of N on the response of the vegetation to $CO_2$, while a strong response to N fertilisation could indicate insufficient N supply and as a result a strong N limitation of the $CO_2$ response. We know that response to increased N supply is globally distributed (LeBauer and Treseder, 2008) and that
$C_3$ plants, which make up the majority of vegetation worldwide, have a positive photosynthetic response to additional atmospheric $CO_2$ (Ainsworth and Long, 2005). However, there is evidence that the +CO2 response would be limited by N availability (forest NPP response to additional atmospheric $CO_2$ is limited by N availability (Norby et al., 2010)) and it is currently unknown whether +N would be similarly affected.

All the models are consistent with the hypothesis of either N or $CO_2$ fertilisation at grid cell level, but the effect does not
necessarily scale to either the regional or global level. The prevalent grid cell level spatial trend is bimodal, with grid cells either having a strong sensitivity to +N or +CO2, but not both (see Fig. 4). Comparing percent change emphasises the dichotomy of +N and +CO2 effects, with most values clustered near either zero for +N or zero for +CO2, but SI Fig. 6 shows that there is no positive relationship or heterogeneous distribution in the absolute values either. The bias toward +CO2 is clear for JSBACH and JULES-ES, with most values varying in +CO2 sensitivity but not +N (Fig. 4, also seen in the
absolute anomalies in SI Fig. 6). A slight tendency towards the reverse is true for CLM4.5, CLM5, and LPJ-GUESS, with more points having a strong +N response and a weaker +CO2 response (Fig. 4). Altogether, LPJ-GUESS and CLM5 show the most areas with both +N and +CO2 sensitivity. Wieder et al., (2019) found that there was a trade-off between +N impact and +CO2 impact in CLM4, CLM4.5 and CLM5, and this seems to be true for our ensemble of models too.

The latitudinal distribution of response shows similarities across models, with high latitudes (shown in purple in Fig. 4) generally more +N sensitive, and the mid latitudes (red to orange on Fig. 4) more +CO2 sensitive. While negative NPP values are present in both +N and +CO2 simulations they occur in different places, with negative NPP occurring in hot arid areas for +N and cold arid areas for +CO2 (Fig. 2, 3, and 4). In hot arid areas +N increases simulates GPP and plant growth but also plant respiration, which then exceed the additional productivity, giving a decrease in NPP. Such model behaviour has been noted before (Meyerholt et al., 2020), however, there is little evidence that such a process would occur in nature. The negative values in all models except CLM4.5 also appear to have a regional bias, with a small number of grid cells responding negatively to both +CO2 and +N in CLM5, JSBACH, and JULES-ES in the subtropics and a larger number of negative values in the subtropics in LPJ-GUESS (Fig. 4). These arid areas appear to be sensitive to neither +N nor +CO2, probably due to low water availability.

We can gain further insights by considering the relationship between responses to +CO2 and +N by forest biome (Fig. 5). The ideal for the models is to be in the area where the observations for +N and +CO2 intersect. Two of the models achieve this partially, JSBACH and CLM5, by having tightly clustered forest vegetation C (VegC) response to +N and forest NPP response to +CO2. The dichotomy between +N and +CO2 NPP response is averaged out at this scale and the models show little of the L-shaped relationship between the +N response and +CO2 response seen at the grid cell level (Fig. 4 and 5).

According to collated N addition experiments we would expect models to have biome level variation in +N response (LeBauer and Treseder, 2008; Schulte-Uebbing and Vries, 2018). Schulte-Uebbing and Vries, (2018) show that tropical forest +N VegC response is lowest and boreal and temperate forest response higher (Fig. 5). While LPJ-GUESS and CLM4.5 capture some variation between averaged biomes, none of the models have the biome responses in the correct order (Fig. 5). However, all the models except LPJ-GUESS tend toward a lower (tropical) +N response. LPJ-GUESS, however, is the only model to have the boreal +N response in the correct range. It is the boreal response that seems to be the main issue, as relative to both the temperate and tropical regions most models show the boreal response as being lower, whereas most of the models have the correct relative +N response for the tropics and temperate regions. Therefore, although the global values of model response are acceptable, the relative spatial patterns show limitations in the reliability of all the models.

## 3.5 N Budget Responses to +N and +CO2

The models' responses in different components of the N budget reflect and affect their overall N sensitivity (Fig. 6). N inputs of BNF and N deposition and loss (we only consider the sum of leaching and gaseous loss so as to be consistent between models) are similar between all the models in the Control simulation (Fig. 6a). The uptake of N by vegetation varies more strongly between models, reflecting differing levels of N mineralisation and assumed N requirements for growth, as also reflected by the different amounts of C and N pools depicted in Fig. 1.

Changes in the N budget components to +CO2 and +N (Fig. 6b and 6c) are not straightforwardly related to changes to productivity (Fig. 2 and 3). For instance, the weak response of NPP to +CO2 in CLM4.5 would suggest only small changes

in uptake compared to the other models (Fig. 2 and 6). However, the +CO2 induced changes in uptake CLM4.5 are higher than that of LPJ-GUESS (Fig. 6b). Similarly, CLM5 has the largest increase in N balance for +CO2 (Fig. 6b) amongst the models, but this does not correspond to a larger response of NPP (Fig. 2f) or uptake response to elevated $CO_2$ (Fig. 6b).

Nevertheless, Fig. 6b reveals a number of important characteristics of the N cycle response to +CO2 underlying the NPP response presented in Section 3.2. For all models except CLM5, which shows a strong response of BNF to elevated $CO_2$, reduced N losses are an important reason for the increased N balance of the ecosystem, which facilitates an increase in NPP in the absence of changes in ecosystem stoichiometry. For all models except CLM5, plant N uptake under elevated $CO_2$ is more enhanced than the change in the N balance of the ecosystem, implying a net transfer of N from the soil to vegetation.

Conversely, the N uptake changes in JULES-ES and JSBACH reflect their sensitivity of productivity to +N and +CO2 (Figs. 2,3, and 6). For JULES-ES we can see that this is driven by changes in loss, particularly for +N, which leads to a much smaller increase in N balance in JULES-ES than the other models. In common with all the models, in JULES-ES the N loss term is a fixed fraction of the mineralisation flux and the soil N pool size. However, JSBACH has less than half the increase in N loss of JULES-ES in the +N simulation (Fig. 6c), low changes in BNF compared to other models (Fig. 7b) and almost

no change in NUE (Fig. 7d). This suggests that in both JULES-ES and JSBACH there is effectively little unmet N demand in the Control scenario.

BNF responses to +CO2 in the models differ in magnitude (Fig. 7a) and mostly are smaller than a meta-analysis of $CO_2$ manipulation suggests (Liang et al., 2016). Only JULES-ES' responses at the global scale and CLM5's boreal response are within the range of the meta-analysis of observations. CLM5 is a clear outlier, with a large increase in BNF. CLM5 takes a C

cost approach to BNF, which is different to the other models (Table 1), and BNF can be acquired for a relatively fixed amount of C (Houlton et al., 2008) and thus when C availability increases under +CO2 the BNF in CLM5 increases. Fisher et al., (2018) conducted a parameter sensitivity analysis of both +CO2 and +N fertilization, which illustrates that both responses are sensitive to the maximum fraction of C from NPP which is available for fixation (a proxy for the fraction of N fixing plants and their efficiency). However, the correct parametrisation of this fraction of C available for fixation is not well

known and further field studies are required. The BNF +CO2 response in the other four models is determined by their simple empirical BNF equations (see Table 1) based on NPP or evapotranspiration. However, new analysis suggests that simple empirical relationships cannot well represent BNF (Davies-Barnard and Friedlingstein, 2020).

The models' BNF response to +N shows one of two responses: a small increase in JULES-ES, CLM4.5, and JSBACH; or a large decrease in CLM5 and LPJ-GUESS (Fig. 7b). The latter models capture the correct BNF sign of response to +N of a

decrease according to the meta-analysis of Zheng et al., (2019), though the amplitude is too large. The former models estimate BNF as a function of NPP resulting in increased BNF whatever the source of the additional NPP is and even when there is sufficient N. Observational evidence (Zheng et al., 2019) shows BNF reduces when N is supplied from another source and it is understood this is because facultative (able to modulate) BNF reduces and obligate BNF is out-competed (Menge et al., 2009). Overall, there is little evidence for any of the BNF functions performing well, primarily due to lack of

robust model parameterisations and parameter values.

The NUE responses allow comparison between models, though comparisons with observations are limited by a lack of field studies. All models have an increase in NUE with +CO2 in line with the current theory of Walker et al., (2015), with the exception of JULES-ES in the boreal region (Fig. 7c). It's unclear why the boreal region is responding differently to both other regions in JULES-ES and other models but the boreal region reduction in NUE under +CO2 likely indicates excess N from mineralisation, possibly triggered by the combination of soil warming and increased atmospheric $CO_2$. CLM4.5 has low NUE response to +CO2 due to fixed C:N ratios, which allow little change in NUE. The other models allow either more allocation to wood or flexible C:N that results in the larger increases of NUE.

There is regional variation in models' NUE response to +N between biomes but all the models in our ensemble reduce NUE in response to +N (Fig. 7d). CLM5 and LPJ-GUESS are distinct in their larger NUE response to +N compared to the other models, but do not share the same geographical spread of response. There is little consistency between models as to which regions have the largest change in NUE. CLM5 has the largest NUE change in the temperate region, whereas in JULES it occurs in the boreal region. No empirical measurements are currently available for NUE response to +N. On the basis that scarcity encourages more frugal use of scarce resource a hypothesis could be that NUE could decrease with increased N availability, as the models show. However, water-use efficiency suggests an alternative hypothesis, as it tends to reduce during drought (Yu et al., 2017). Overall, the large variations in signal and sign of BNF and NUE responses to +N treatment between models suggest there is considerable uncertainty in our understanding.

## 4 Discussion

In this paper, we investigated the performance of five nitrogen-enabled land surface models that are part of current generation Earth System Models used in the framework of CMIP6 (Eyring et al., 2016). These new N-enabled land surface models in CMIP6 reproduce key global carbon cycle metrics. Despite the importance of N availability for regional productivity, there is large and unconstrained uncertainty in the magnitude of the global and regional N fluxes (Fig. 1).

We have focused on three general components of N-enabled models that affect the plant N uptake and eventual productivity: N inputs via BNF; NUE; and the N losses. We find that all three show considerable heterogeneity of response between models. Previous studies suggest that stoichiometric controls and the processing of soil organic matter are important for a realistic +CO2 response (Zaehle et al., 2014a). These are essentially contributory factors to NUE, where we find large variation between models (Fig. 7). The lack of well-constrained observations for global and biome-level NUE and N loss responses implies that these areas need more work. N loss is particularly challenging, as there are multiple pathways (leaching, flooding, gaseous loss, fire, land use change, etc.) and forms ($N_2O$, $N_2$, etc.) of loss and each model represents these in different ways. More observational studies and syntheses of existing observations are needed to quantify the nitrogen cycle in different biomes. In particular, better constraints are needed for the N cycle response to perturbations.

All the models show a global average productivity response to increased atmospheric $CO_2$ commensurate with those recorded in field studies. However, the regional responses and mechanisms behind this response vary widely, resulting from

the interaction of the instantaneous physiological response to elevated $CO_2$ (e.g. Ainsworth and Long, (2005), which is embedded in all five models (but see Rogers et al., (2017)), with limitations imposed by temperature, water, light, and nitrogen, as well as the response-time of vegetation dynamics. For instance, in LPJ-GUESS and CLM5 the response to elevated $CO_2$ in semi-arid tropical ecosystems is smaller than that of temperate ecosystems or other models. This suggests a combined effect of water and nitrogen limitation on soil organic matter decomposition in these models, and thus low nitrogen availability that is not compensated for by changes in BNF. Similarly, tundra and arctic responses to elevated $CO_2$ varies widely across the models and is associated with the representation of BNF. This large regional variance highlights the need for more comprehensive observational data to constrain responses to elevated $CO_2$, particularly in under-sampled regions such as the high arctic and tropical semi-arid regions (Song et al., 2019).

The growth response to N addition across models is more varied. Two of the five models (JULES-ES and JSBACH) have little productivity response to increased N availability, indicating that they do not have any significant limitation of the C cycle by N availability (Fig. 3). There are four substantial similarities between these two models (Table 1): (i) the use of NPP to determine BNF; (ii) a direct control of NPP by N availability, whereas photosynthetic C uptake (GPP) is not directly affected by N (Goll et al., 2017; Wiltshire et al., 2020); (iii) the use of dynamic (as opposed to prescribed) vegetation, where vegetation cover is determined by the climate input to the model; and (iv) the assumption that N availability in pre-industrial times was sufficient to sustain the C cycle everywhere on land because observed present-day N limitation is a result of anthropogenic changes, most notably increased $CO_2$ (Goll et al., 2017).

The hypothesis of no pre-industrial N limitation is based on the assumption that prior to industrial times, the conditions of natural terrestrial ecosystems were stable for sufficient time to permit any lack of N availability to be filled by biological nitrogen fixation (Thomas et al., 2015). Consequently, the pre-industrial Control run with both N and C is very similar to the C-cycle only version and a C equilibrium is reached before a N equilibrium. The disjoint between the C and N equilibriums may lead to varying levels of simulated N availability and may affect the model responses to perturbations. While there is evidence for wide-spread (co-) limitation of NPP in recent decades (LeBauer and Treseder, 2008; Song et al., 2019; Vitousek and Howarth, 1991), there is insufficient data to test the hypothesis of no pre-industrial N limitation. A summary by Thomas et al., (2015) suggests reasons that pre-industrial productivity of terrestrial ecosystems was affected by ecosystem N availability, e.g. the presence of unavoidable losses to denitrification, or the competitive exclusion of nitrogen fixing species as ecosystems mature. The inability of JULES-ES and JSBACH, when initialised in the assumption that pre-industrial N availability does not limit vegetation growth, to simulate observed N addition responses comparable to models without this assumption suggests that this may be an important component of the N cycle constraint on the global C cycle. No pre-industrial N limitation also drives other model decisions (such as N limitation not being incorporated into the GPP equation, see Table 1), which may further contribute to the models being under-sensitive to N compared to observations.

The models mostly represent changes in productivity from +N in high latitude northern hemisphere regions less well than other parts of the world as a percentage, as covered in the results section 3.3, Fig. 5, and SI Table 3. While the low NPP of these regions make them more likely to have high percentage increases, the mean Polar +N response across the models is 8 –

59%, broadly in the range of a meta-analysis of observations 12 – 64% (LeBauer and Treseder, 2008). But looking at the maps of response (Fig. 3), the model response is either too low or too high compared to the aforementioned observational range. High latitude tundra is an important but difficult to model biome because of the potential for release of methane (Nauta et al., 2015), permafrost C and N release (Anisimov, 2007; Burke et al., 2012; O'Connor et al., 2010), albedo changes with vegetation expansion (Myers-Smith et al., 2011) and the difficulty in representing large amounts of C stored in soil. This complexity in C and N cycle is not always well understood or represented in models and therefore could limit the ability of models to provide accurate responses to perturbation. A fully integrated model that accounts correctly for all of these is not yet possible but is necessary to reduce uncertainties.

The greater Amazon basin is a critical area of interest for the future of the terrestrial carbon balance under climate change. Our simulations show that for most models, NPP in this area increases with +CO2, but all the models find a small or no change in NPP with +N. These regions are thought to be phosphorus rather than N limited, due to depletion through weathering over long periods. This result supports the idea that favourable climate conditions cause a high leaf area index (LAI) in this part of the tropics, such that there is little margin for increased NPP from +N (Fisher et al., 2018). For +CO2 there is the potential for increased NPP because of either increase in NUE or decreases in N losses, giving productivity increase without an increase in LAI. Reducing the uncertainty in NPP response to +CO2 is important, as the moist tropics represent a significant proportion of the world's aboveground biomass and therefore the size of the overall terrestrial sink will be influenced by the $CO_2$ uptake in this biome.

This experimental setup considers +N and +CO2 separately, but not the combined effects. It cannot be assumed that the effect of both +N and +CO2 on productivity are linearly additive. It has been shown elsewhere that LPJ-GUESS (Wårlind et al., 2014) and BIOME-BGC (Churkina et al., 2009) have a significant non-linear (synergetic) term between $CO_2$ and N deposition. An assessment of the combined effects of +N and +CO2 may show a significantly different picture of model performance.

Part of the uncertainty in the models comes from the reanalysis climate dataset used to drive the models. CRU-NCEP was chosen for the good spatial and temporal coverage, but some biases exist in the data compared to climatologies such as WATCH (Weedon et al., 2011). Offline simulations driven by low forcing frequency (six-hourly) CRU-NCEP data significantly overestimate evapotranspiration in regions with convective rainfall types and thereby could affect stomatal conductance and photosynthesis (Fan et al., 2019). Responses to +N and +CO2 may partially be shaped by other limiting factors such as water availability, which will be handled differently between models, limiting the insight on the exact processes that control model responses to change. This does not affect all the models equally, as some are known to be sensitive to the driving climatology. JSBACH, JULES-ES and LPJ-GUESS may be particularly strongly affected due to their dynamic vegetation. Lawrence et al., (2019) show that CLM5 corresponds best to benchmarks with GSWP3 forcing dataset (Hurk et al., 2016) and work with JULES shows that climate forcing is the biggest cause of variance of those considered (Ménard et al., 2015).

As well as uncertainty in the models, the observational data also has uncertainties and limitations. Global benchmarks are approximate measures, as multi-faceted process mechanics are integrated over large domains and generalized, e.g., over climate zones that are inherently variable. Of the limited global or regional observations available, many use interpolation or proxies such as satellite data to upscale relatively small amounts of direct observational data. The perturbed responses may also have uncertainties beyond the spread of the observed responses because of the small observation basis and potential

biases in the geographical sampling. Therefore, they may suffer from leverage points and skew in the data towards more accessible, higher income, or higher population areas, such as western Europe, which are not representative of where models are impacted most at the global scale. One of the +N global responses cited is based on 126 values from LeBauer and Treseder, (2008) but may over-estimate the global response by including high responses from young tropical soils. The NPP response to +CO2 response for woody plants total above ground biomass (Fig. 5) is based on just 16 experiments (Baig et

al., 2015), making the upscaling to biome scale less reliable than if more data were available. These meta-analyses combine measurements from a range of time periods and places, and different conditions (e.g. gradual or instantaneous perturbations) and thus models run at a global scale cannot be expected to be entirely consistent. Hence statements about the marginal issues of model accuracy are unlikely to be robust as further observational constraints may alter the perspective.

## 5 Conclusions

This is the first systematic comparison of the responses to increased N (+N) and $CO_2$ (+CO2) in LSMs with terrestrial N cycles contributing to CMIP6. The five models considered here yield fair overall agreement with global and tropical observations but are less robust in high latitude regions.

   The models are not equally sensitive to either +CO2 or +N, with individual grid cells tending to respond to either +N or +CO2. However, at the regional and global scale this pattern is averaged away and there is little correlation. Within this

ensemble there is clear distinction between models that show strong N limitation, e.g. CLM4.5, which has a low NPP response to +CO2, and models that show very weak N limitation, e.g. JULES-ES and JSBACH, which have a low NPP response to +N. The two models with intermediate N limitation (CLM5 and LPJ-GUESS) capture the global scale response to +CO2 and +N reasonably well. However, although CLM5 performs well by many metrics, it is an outlier compared to other models or observations as its BNF and the NUE response to $CO_2$ appears to be larger than supported by observations.

Similarly, LPJ-GUESS captures NPP responses to +CO2 and +N well at the global level but overestimates the vegetation C response to +N in forested tropical and temperate biomes.

   The model initialisation with or without the assumption of sufficient N in pre-industrial times is a key determinant of the differences between the models. The presence of N limitation before the rise of atmospheric $CO_2$ levels is an important and challenging question to resolve. While further modern constraints on +N response may inform which approach is more

realistic, understanding from reconstructions or other data sources could help resolve this question.

The wide range of empirical or semi-mechanistic representations for key processes such as BNF, NUE, and N loss, show how important further process understanding is for many parts of the N cycle. These parts of the models are influential, but because N cycle components are a recent addition to LSMs, fewer data are available to evaluate N cycle processes than for C cycle components. The addition of this representation of N limitation on C uptake is a big step forward in this generation of models, addressing the biggest systematic bias in future projections of land C sinks. However, it is now crucial to better constrain their behaviour at regional and process levels. Consequently, better observational constraints are required to understand whether models are working appropriately, even when the process understanding is improved.

## Acknowledgements

The research data supporting this publication are openly available from the University of Exeter's institutional repository at: https://doi.org/10.24378/exe.2624

Authors acknowledge the work by Dr Johannes Meyerholt (deceased 2020) that formed the basis of this paper.

Authors acknowledge funding from the European Union's Horizon 2020 research and innovation programme under grant agreement No. 641816 Coordinated Research in Earth Systems and Climate: Experiments kNowledge, Dissemination and Outreach (CRESCENDO).

SZ acknowledges support by the European Union's Horizon 2020 research and innovation programme under grant agreement No. 647204 (QUINCY).

VB, PF and SZ acknowledge funding from the European Union's Horizon 2020 research and innovation programme under grant agreement No. 821003 (4C project).

RF was supported by the National Center for Atmospheric Research, which is a major facility sponsored by the NSF under Cooperative Agreement 1852977.

BS and DW acknowledge this study is a contribution to the Strategic Research Area MERGE and the Swedish national strategic e-science research program eSSENCE.

CDJ & AJW were supported by the Joint UK BEIS/Defra Met Office Hadley Centre Climate Programme (GA01101).

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

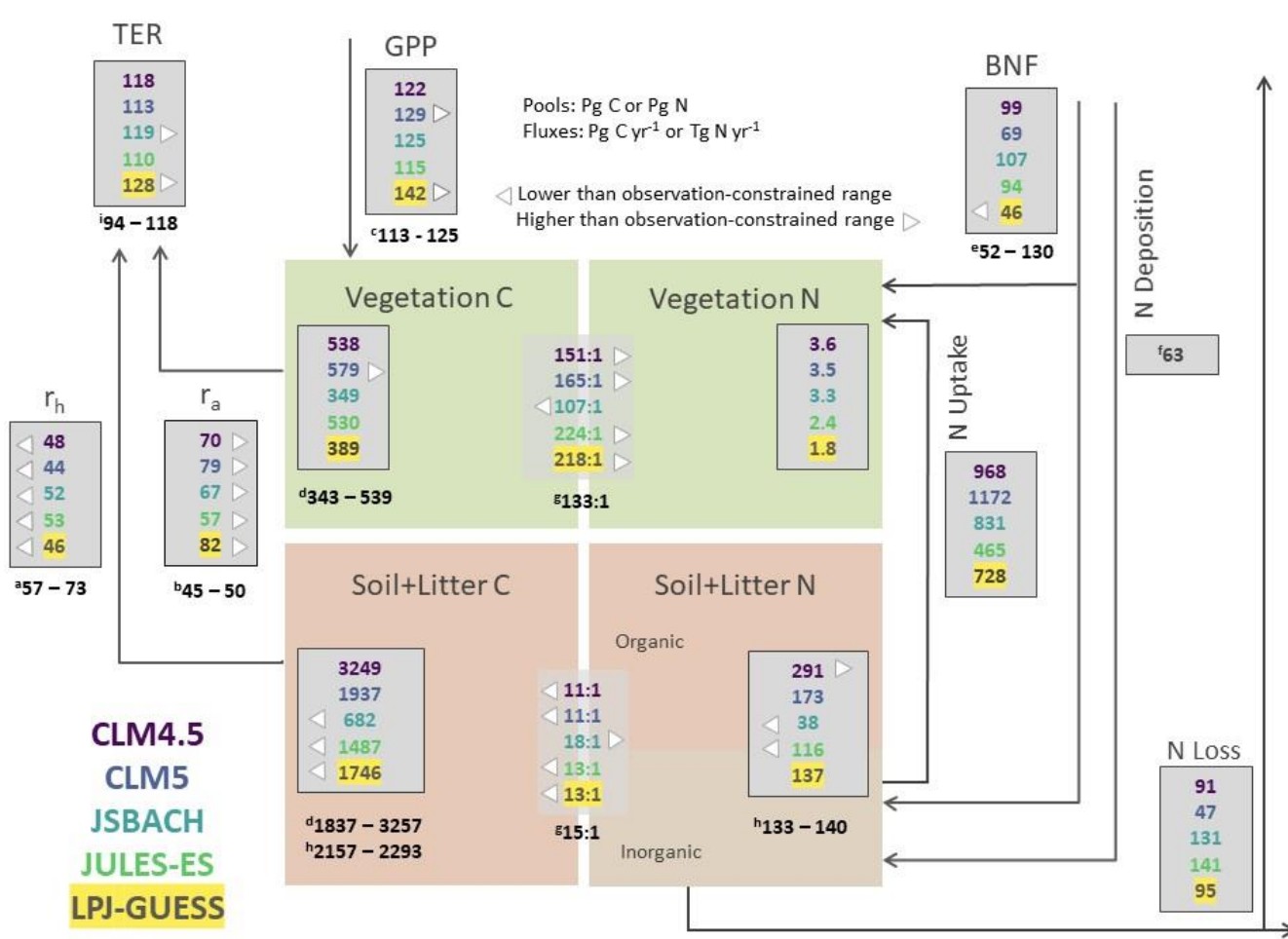

**Figure 1. 1996-2005 mean model estimates of the major ecosystem C and N component pools and fluxes in comparison with observation-based estimates from the literature. C = Carbon; N = Nitrogen; rh = Heterotrophic respiration; ra = Autotrophic respiration; GPP = Gross primary productivity; SOM = Soil organic matter; BNF = Biological nitrogen fixation; The N uptake flux refers to root uptake of inorganic N. Ranges shown represent the 95% confidence intervals, standard deviation, or similar uncertainty metrics, where available. Where observation-based ranges or values are available an arrow indicates that either the model value is higher than the range or lower. Where there is no arrow, the model is within the observation-based range or there is no observation-based range to compare to. N loss is the loss via gaseous loss and leaching. The black numbers indicate observation-based estimates from the literature: a) Heterotrophic respiration: Bond-Lamberty and Thomson, (2010), soil respiration estimate for 2008. To account for the included root respiration, we reduced the literature estimate by 33% according to (Bowden et al., 1993); b) Autotrophic respiration: Piao et al., (2010), Luyssaert et al., (2007), present day estimate for forests from 2007; c) GPP: Jung et al., (2011), averaged estimate for 1982-2011; d) SOM+Litter, and Vegetation C: Carvalhais et al., (2014), present day estimate from 2014; e) BNF: (Davies-Barnard and Friedlingstein, 2020) upscaled averages for 1980-2019; f) N deposition: (Lamarque et al., 2013), estimate for 2000; g) C:N ratios for soil and vegetation: Wang et al., (2018); h) Soil nitrogen in the top 1 meter and soil carbon in the top 1 meter (Batjes, 2014); i) Total Ecosystem Respiration: (Ballantyne et al., 2017).**

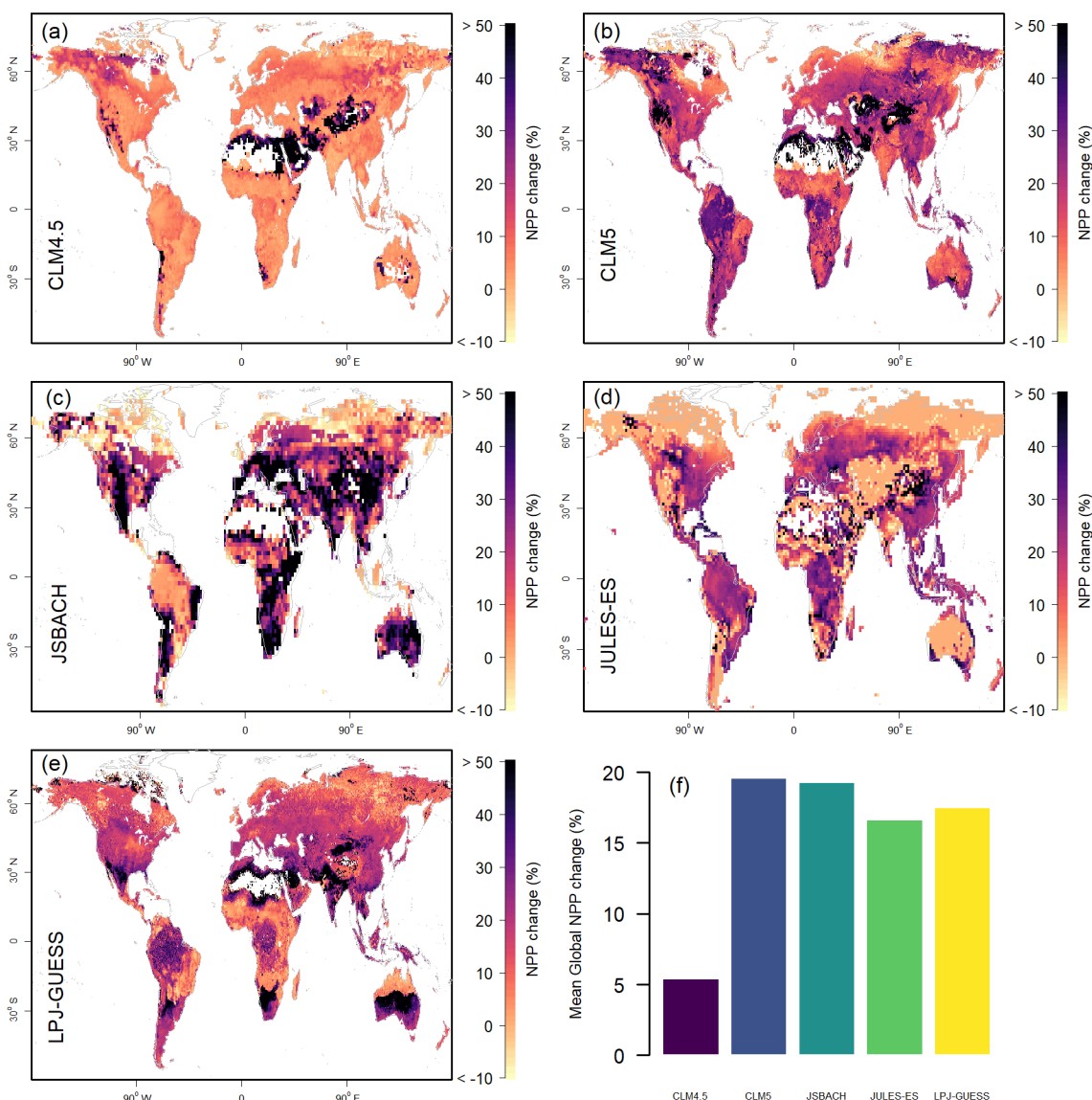

**Figure 2. Model estimates of 1996 - 2005 mean net primary productivity (NPP) response to +CO2. (a) – (e) Model estimates, shown as the anomaly compared to the model control scenario. Values above 50% are given the 50% colour. (f) Global percent change in mean NPP.**


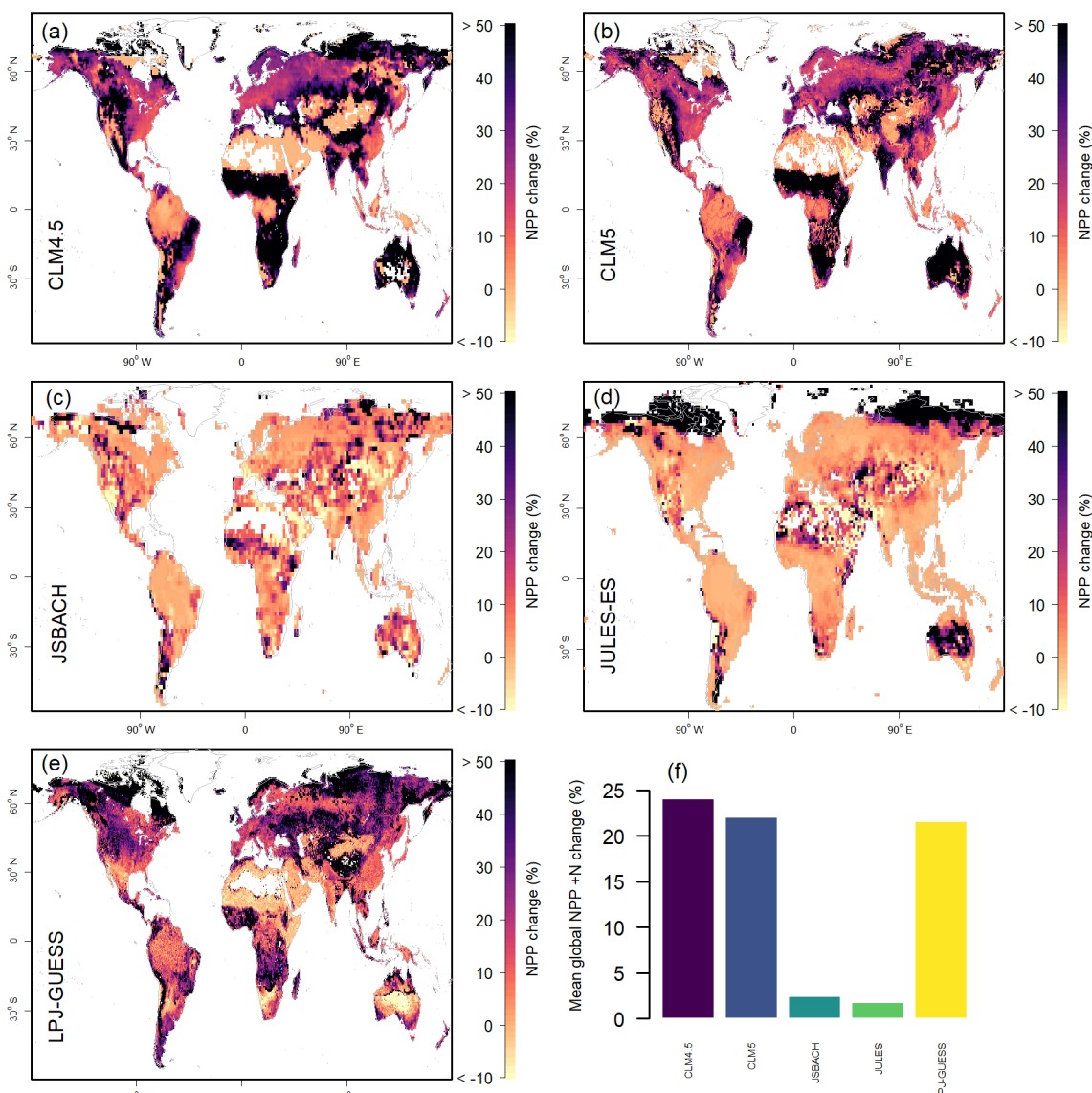

**Figure 3. Model estimates of 1996-2005 mean net primary productivity (NPP) response to +N. (a) – (e) Model estimates, shown as the anomaly compared to the model control scenario. Values above 50% are given the 50% colour. (f) Globally integrated values. Global percent change in mean NPP.**

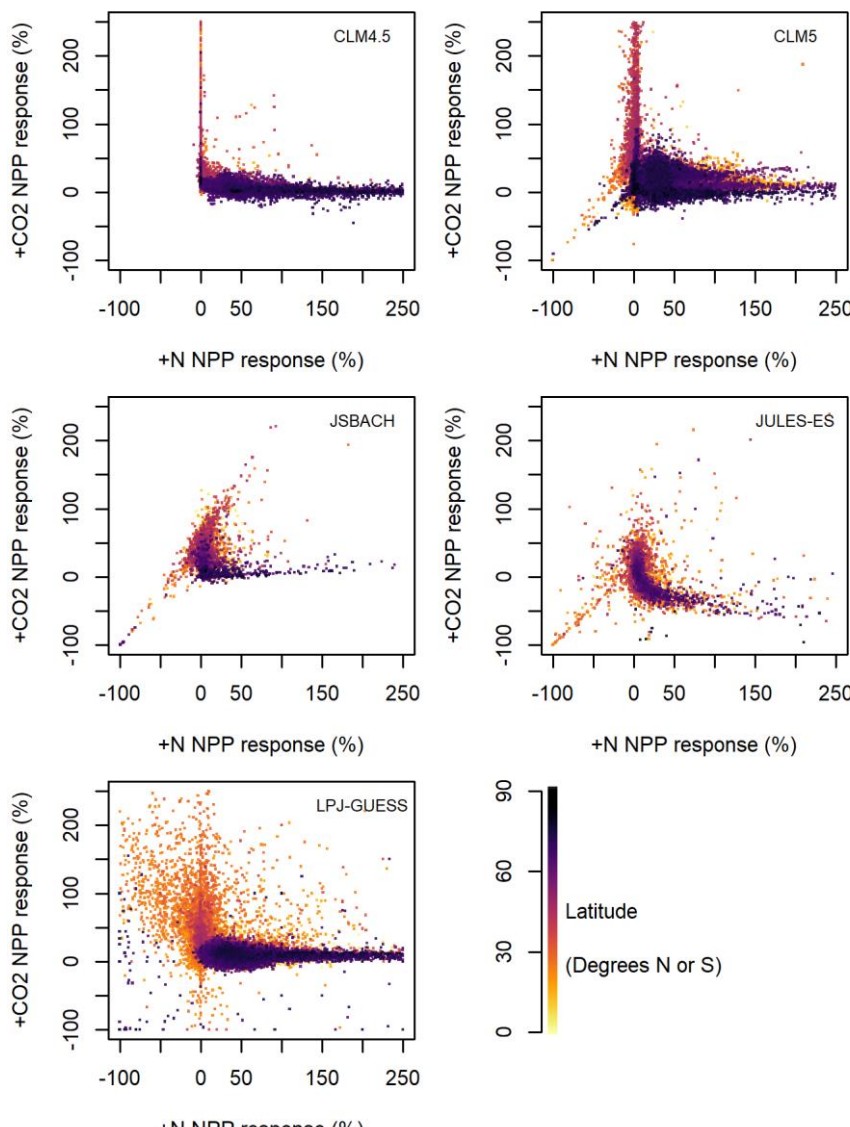

**Figure 4. Model estimates of 1996-2005 mean net primary productivity (NPP) response to +N vs +CO2, as a percent anomaly of the control scenario. Each grid box is plotted against the corresponding grid box for the other variable. The percent change is capped at 250% and values above are not plotted. The colour of the points indicates the latitude either North or South.**


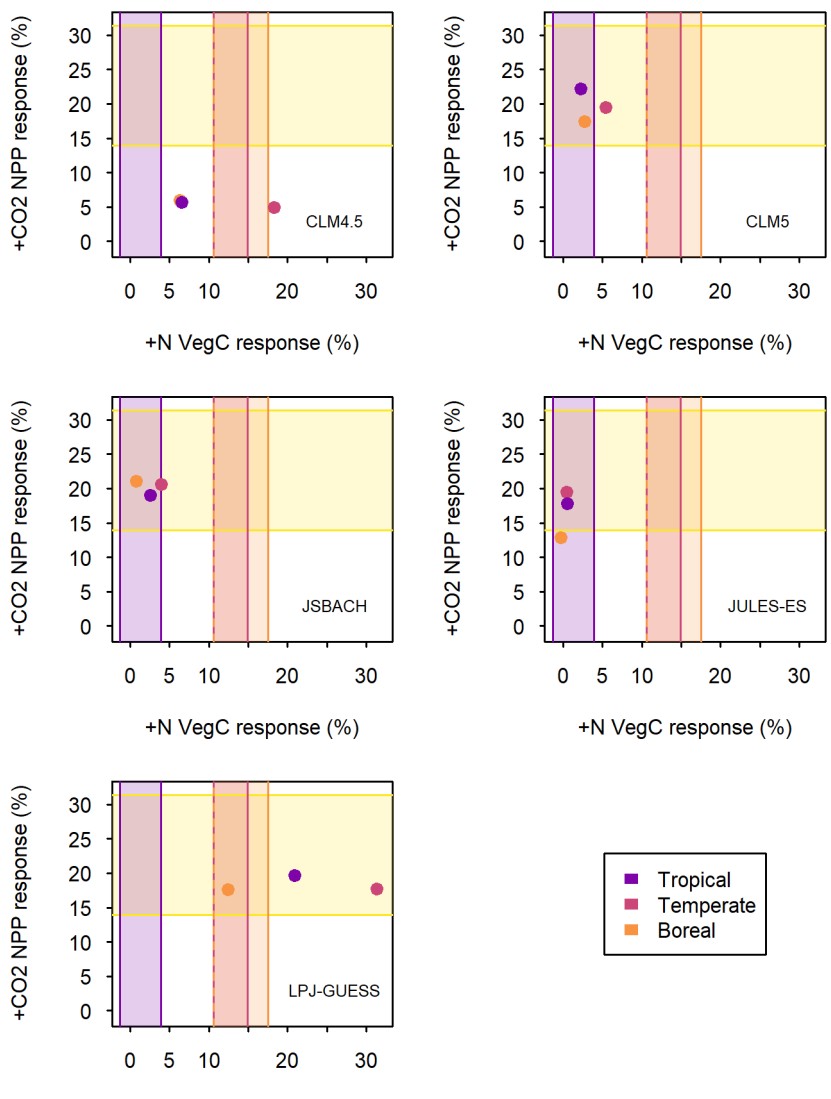

**Figure 5. Average 1996-2005 model predictions of woody plant NPP responses to +CO2 (y-axis) and aboveground forest vegetation C pool size responses to nitrogen (N) addition (x-axis) for each of the models (as labelled). Area outlined in yellow indicates synthesis of observed woody plant NPP responses to +CO2 (Baig et al., 2015). Other coloured areas indicate biome-wise estimates of aboveground forest C change per added N (Schulte-Uebbing and Vries, 2018). For +CO2, NPP is restricted to simulated vegetation with NPP > 0.2 kg C m⁻² yr⁻¹ to exclude non-forest areas. For +N, forest VegC in CLM5, CLM4.5, and LPJ-GUESS is taken from wood C and N, whereas all C and N is included for JULES-ES and JSBACH due to model output limitations. The biomes are allocated according to Köppen-Geiger climate classification (Kottek et al., 2006). The lower limits for Temperate and Boreal +N are the same value.**



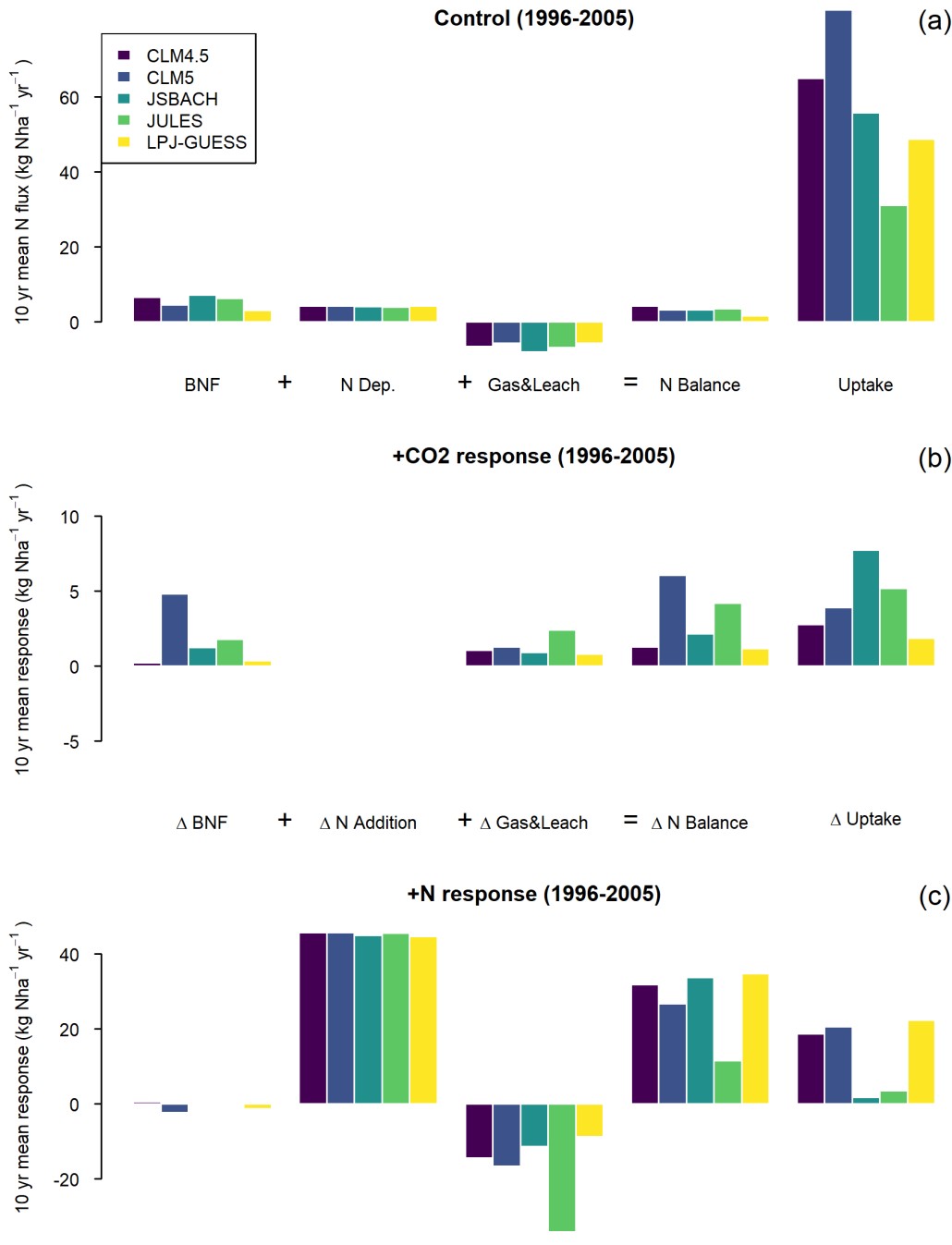

**Figure 6. Global averaged 1996-2005 biological nitrogen fixation (BNF), N deposition, N loss via gaseous and leaching, the balance**
**of those three inputs/losses, and the plant N uptake of the models. The top panel represents the Control scenario, and the second and third panels the response to +CO2 and +N perturbations (see methods). Note that the y-axis scale is 4x smaller for +CO2 response than the Control or +N response. All changes are relative to a nominal N pool in the terrestrial biosphere. Gas and**

**Leaching loss is therefore shown as a negative (a loss from that N pool) in the Control. In the +CO2 and +N responses a positive change in Gas&Leach indicates less losses than in the Control scenario, and a negative change more losses than the Control.**


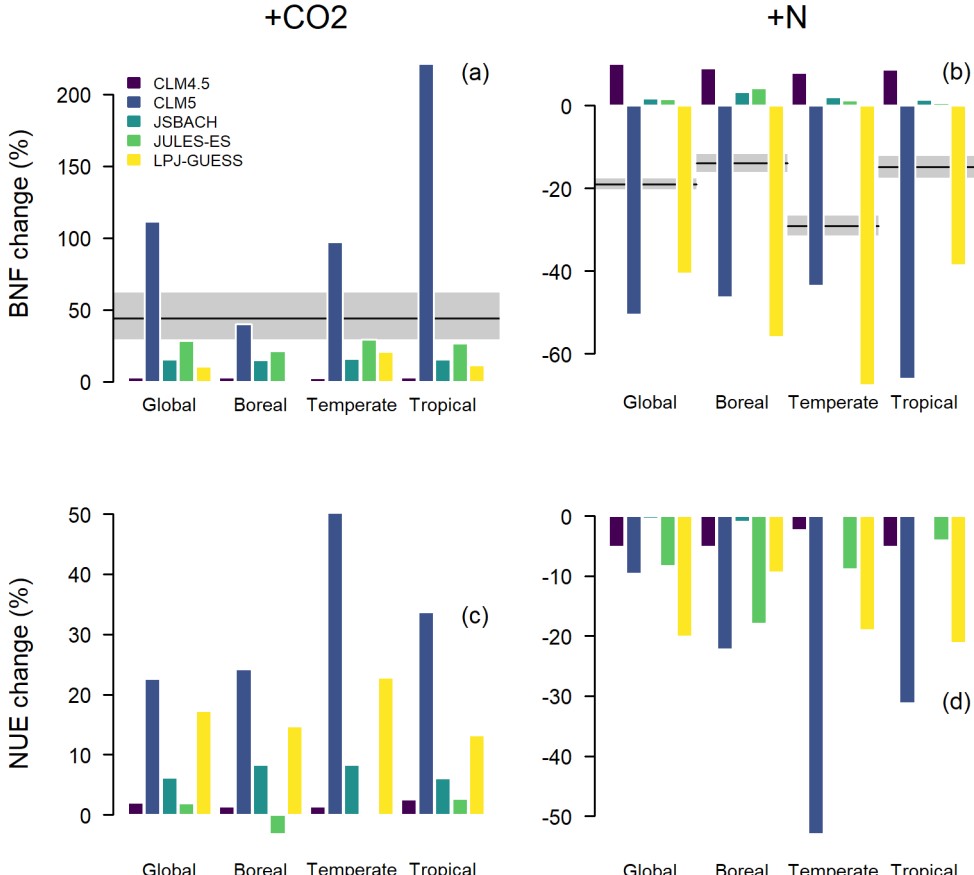


**Figure 7. Averaged 1996-2005 responses in biological nitrogen fixation (BNF) and nitrogen-use efficiency (NUE; see Eq. 1) to +CO2 and +N perturbations for the global (all vegetation types) or forest region averages. (a) Model BNF responses to +CO2. Black line and grey area indicate mean and 95% CI of the global estimate published by Liang et al., (2016). (b) Model BNF responses to +N. Black lines and grey areas indicate means and 95% confidence intervals of the forest estimates published by**
**Zheng et al., (2019). (c) Model NUE responses to +CO2. (d) Model NUE responses to +N. Forest biomes are according to Köppen-Geiger climate classification (Kottek et al., 2006), see SI Fig. 1.**

**Table 1. Key nitrogen cycle algorithms applied by the models. C = Carbon; N = Nitrogen; GPP = gross primary productivity; NPP = net primary productivity; PFT = plant functional type.**

| | CLM4.5 | CLM5 | JSBACH | JULES-ES | LPJ-GUESS |
|---|---|---|---|---|---|
| **Key references** | Oleson et al. (2013) | Lawrence et al. (2020) | Goll et al. (2017), Mauritsen et al. (2019) | Wiltshire et al. (2020) | Smith et al. (2014) |
| **N effect on GPP** | Downregulation of GPP to match stoichiometric constraint from allocable N | Leaf N compartmentalized into different pools to co-regulate photosynthesis according to the LUNA model | No direct effect | No direct effect | Reduction of rubisco capacity in case of N stress |
| **N effect on autotrophic respiration** | N content-dependent tissue-level maintenance respiration | Updated PFT-specific N-dependent leaf respiration scheme | No direct effect | N content-dependent maintenance respiration for roots and stems | N content-dependent maintenance respiration for roots and stems; leaf respiration reduced under N stress |
| **Vegetation pool C:N stoichiometry** | Fixed for all pools | Flexible for all pools | Fixed for all pools except labile | Flexible leaf stoichiometry from which root and stem C:N are scaled with fixed fractions | Flexible for leaves and fine roots, fixed otherwise |
| **Retranslocation of N from shed leaves** | Fraction of leaf N moved to mobile plant N pool prior to shedding. Fraction depends on PFT-specific fixed live leaf and leaf litter C:N ratios. | Fraction of leaf N moved to mobile plant N prior to shedding via two pathways: a free retranslocation, or a paid-for retranslocation dependent on PFT- | Fraction of leaf N moved to mobile plant N pool prior to shedding | Fraction of leaf N moved to labile store with PFT specific retranslocation coefficient | Fraction of leaf N moved to mobile plant N pool prior to shedding. Fraction depends on N stress. |

| | | | | | |
|---|---|---|---|---|---|
| | | specific dynamic leaf C:N range and minimum leaf litter C:N and available carbon to spend for extraction in FUN model | | | |
| **Biological N fixation** | Monotonically increasing function of NPP | Symbiotic N fixation according to the FUN model, asymbiotic N fixation linearly dependent on evapotranspiration | Non-linear function of NPP | Linear function of NPP, 0.0016 kg N per kg C NPP | Linear function of ecosystem evapotranspiration, 0.102 cm yr$^{-1}$ ET +0.524 per kg N ha$^{-1}$ |
| **Ecosystem N loss** | Denitrification loss as fraction of gross N mineralization + fraction of soil inorganic N pool in case of N saturation (CLM-CN) / Denitrification as fraction of nitrification (CENTURY) Leaching as function of soil inorganic N pool size Fractional fire loss as fraction of vegetation and litter pools | Denitrification as fraction of nitrification (CENTURY) Leaching as function of soil inorganic N pool size Fractional fire loss as fraction of vegetation and litter pools | Denitrification proportional to soil inorganic N pool and soil moisture Leaching proportional to soil inorganic N pool and drainage | Denitrification is a fixed fraction (1%) of mineralization flux Leaching of nitrogen is a function of soil inorganic N pool, drainage, and a parameter representing the effective solubility of nitrogen | Denitrification as fixed fraction of mineralization flux Leaching as function of soil inorganic N pool and drainage N loss from fire events |
| **Plant N uptake** | Function of plant N demand, soil inorganic N | Soil uptake of inorganic N according to the | Plant N demand-based, limited by soil inorganic N | Demand based on GPP and limited by soil inorganic N | Determined to maintain optimal leaf N for photosynthesis, limited |

| | | | | | by soil inorganic N availability, fine root mass, soil temperature and plant N status |
|---|---|---|---|---|---|
| | availability, and competition with heterotrophs | FUN model | availability | availability | |


Table 2. Percent change in mean global NPP from perturbations. The observations come from meta-analyses which may not be directly comparable, but which provide a useful context.


| | +CO2 | +N |
|---|---|---|
| **CLM4.5** | 5.4% | 24.1% |
| **CLM5** | 19.6% | 22.1% |
| **JSBACH** | 19.3% | 2.5% |
| **JULES-ES** | 16.7% | 1.8% |
| **LPJ-GUESS** | 17.5% | 21.7% |
| **Mean whole plant NPP percent change based on meta-analyses of field scale measurements** | 15.6% (2.8 – 28.4%) (Song et al., 2019) | 6.5% (3 – 10.5%) (Song et al., 2019) |
| **Mean productivity value percent change based on meta-analyses of field scale measurements** | 26% (12.2 – 39.8%) (Song et al., 2019) (ANPP) 22.3% (13.9 – 31.4%) (Baig et al., 2015) (total woody plant biomass) 21.4% (11 – 32.8%) (Baig et al., 2015) (above-ground woody plant biomass) | 20% (7.5 – 32.5%) (Song et al., 2019) (ANPP) 29% (22 -35%) (LeBauer and Treseder, 2008) (ANPP) |