# Peer review of "Nitrogen Cycling in CMIP6 Land Surface Models: Progress and"

_Biogeosciences, 2019_

## Referee Comment (RC1) · Vivek Arora (Referee) · 7 Feb 2020

Authors compare the behaviour of coupled terrestrial N and C cycles in five models that are contributing results to sixth phase of CMIP (CMIP6). The subject of the manuscript is of clear and significant interest to the Earth system modelling community as more and more land components in ESMs explicitly represent terrestrial N cycle and given the large spread among land C cycle models. However, in its current state the manuscript appears to be written hastily with several points unclear, statements that are weakly supported, some incorrect statements, and at places the analysis of results is as simple as which model produces high values of a given quantity and which low.

I have three major comments.

[Figure]

First, nitrogen used efficiency (NUE) as introduced in equation (1) is simply C:N ratio. In the current literature NUE is typically defined as an efficiency indicator for the utilization of nitrogen in agriculture and food systems (Fageria and Baligar, 2005). That is, higher the NUE the lower amount of applied N enters the environment. I suggest, to avoid confusion with existing definition of NUE authors simply use C:N ratio in equation (1).

Second, the authors have compared the results of two experiments, +CO2 and +N, from models with observation-based estimates. I feel that, the observation-based estimates and the experiments they were based on have not been properly introduced. Nor do the authors discuss limitations of these real world experiments whose results are used to evaluate models. For example, the results from +CO2 experiment used to evaluate models are based on the Baig et al. study which is a meta-analysis but a reader is never told about this. How many studies does this meta-analysis summarizes results from? Similarly, for the +N experiment, the LeBauer and Treseder (2008) study is also a meta-analysis. Both these meta-analyses, results from which are used to evaluate models, should be properly introduced and their limitations discussed. For example, the +CO2 type experiments done are based on instantaneous doubling of CO2 while in the real world CO2 is increasing gradually. Similarly, in +N experiments additional N application rates, I think, are increased instantaneously while in the real world N deposition rates have increased gradually. In addition, can the average results from meta-analysis be used to evaluate the globally-averaged response. The photosynthesis theory says that the CO2 fertilization effect must be strongest in the tropics. How does one account for this? Were the studies used in meta-analysis uniformly distributed geographically speaking? As a modeller myself, I realize, the business of evaluating models is difficult but as long as limitations of observation-based estimates are mentioned, it allows readers (and authors too) to make a rationale and informed expectation of the extent to which observations and models should compare well with each other.

My third comment is that as a reader, after reading this manuscript, I am not sure if I

know anything more about N cycling in models than I did before. I feel, the results from these models need to be analyzed and reported in a much more clever way to provide overarching conclusions. Note that the ability of models to simulate recent trends in GPP and NBP is not due to N cycle. Models without N cycle can achieve this too as is seen in the TRENDY intercomparison which contributes results to annual Global Carbon Project studies (Le Quéré et al., 2018).

Other comments

Page 1, abstract, lines 26-28. Upon reading these lines it is clear that 200 ppm CO2 and 50 Kg N/hectare.year N deposition increase are both hypothetical. But as a reader I was wondering what observations are used. At this point in the abstract the reader is not aware that model results are being compared to results from meta-analyses later in the manuscript.

Page 3, line 85. Please consider rewording "All models ran a global spin-up for all ecosystem pools up to the year 1860" to "All models pools were spun up to equilibrium using climate and other forcings corresponding to year 1860".

Page 4, Section 2.2 and 2.3. Please consider summarizing in a table the runs performed. After the pre-industrial spin up, it seems, three runs have been performed – a 1861-2015 historical simulation, a +CO2 simulation for the period 1996-2015, and a +N simulation for the period 1996-2015.

Page 4, equation (1). Please use C:N ratio in this equation as opposed to NUE.

Page 4, equation (2). This equation is incorrect. Change in NPP cannot be simply determined by multiplying the changes in NUE and N uptake. Please see https://en.wikipedia.org/wiki/Product_rule which explains the product rule of differentiation.

Page 4, equation (3). Please define delta N (which implies N balance, I think) properly in words. It seems it is the change in total amount of N in the land (Tg N). But the right

hand side terms of the equation are all fluxes which implies the units of N should be Tg N/year. I am confused. The term "N balance" is used throughout the manuscript. It is an important term and yet in the absence of clear worded definition and units it is difficult to follow the context in the rest of the manuscript where this term is used.

Page 5, lines 136-137 reads "This generation of N models are generally consistent within observational constraints, showing an improvement compared to CMIP5 N models". However, nowhere in the manuscript have model results from CMIP5 models been shown so how can one conclude CMIP6 models are better than CMIP5 models. Please reword this sentence.

Page 6, line 168. Please consider replacing "non-N model structure" with "C cycle related processes".

Page 6, line 174 reads "Across the ensemble there is a slight correlation between the global GPP total and NEP". Please note that for the pre-industrial spin up models' NEP is zero since the model has been spun up to equilibrium. This implies for the pre-industrial state there is no correlation between GPP and NEP. Over the historical period, there is no reason to expect a strong correlation between absolute GPP values and NEP. What is expected is a strong correlation between rate of increase of GPP and NEP since it is the rate at which GPP increases that determines the land C sink.

Page 6, lines 175-176 are unclear.

Page 6, line 186 reads "BNF on the other hand has a wider observed range . . .". For a reader it is unclear where the observed range of BNF comes from.

Page 7, lines 202-204 read "Looking at inputs and losses excluding anthropogenic N addition (BNF + N Deposition – N Loss), all the models have a surplus of N and could be said to be 'open' systems with regard to N balance". I am not sure what this means. Recall that after the pre-industrial spin up the sum of all model input N fluxes should ideally be the same as output model fluxes. Was this evaluated? During

the transient simulation additional N deposition and fertilizer input leads to increased gaseous losses of N, perhaps increased leaching, and accumulation of N in organic and inorganic pools. I am unsure what 'open' and 'surplus of N' means– does it mean all the additional N input is lost as gaseous fluxes and to leaching. We all know BNF (especially due to increase in crop area) and N deposition increase over the historical period so N balance, as defined in the manuscript, will always be +ve. What's more important here is where does this additional N ends up?

Page 7, line 206 reads " ... Soil+Litter C is generally low, compared to observational estimates ...". Does the observation-based estimates contain C from peatlands? The CMIP6 models, I suppose, do not account for C in peatlands and perennially frozen C in permafrost. Could this be the reason for low model estimates.

Page 7, line 209. "Comparing the C:N of Soil+Litter global total weight the ratios are similar across models ...". This sentence doesn't read properly. Also, this section reports the reason for higher C:N ratio of the soil organic matter in the JSBACH model as "The higher ratio for JSBACH is due to the 10:1 ratio for slowly decomposing soil carbon (humus) and larger ratio for litter". I cannot follow what this sentence is trying to imply. This is true for all models. Soil C always decomposes slowly than litter.

Page 7, Section 3.1. In Figure 4, sub panel, it seems the global model response is compared to observation-based estimate from Baig et al. 2015. Is the Baig et al. 2015 average representative of the whole globe or weighted heavily towards certain geographic regions.

Page 8, lines 226-227 read "Therefore, although the models reach a majority consensus on +CO2 NPP effects overall, the important regional details are still contradictory". Does OVERALL in this sentence means globally? When the manuscript says the "important regional details are still contradictory", I think, it is meant that regional response to +CO2 do not agree amongst models. I think, it doesn't mean they the models contradict some observations because there aren't any regionally aggregated

+CO2 observation-based responses. Please reword this sentence.

Page 8, lines 235-239. I wonder, if there a way to quantify or plot this dichotomy between +N and +CO2 responses.

Page 8, lines 255-256. "The largest responses to +N and +CO2 of input and loss do not necessarily correlate with either N uptake or changes to productivity". I am not sure what this sentence means.

Page 9, line 267. "In contrast, JSBACH has less than half the increase in loss of JULES in the +N simulation". By the time, a reader reaches this sentences he/she may forget what quantity is being referred to. Does this sentence refers to plant N uptake?

Page 9, lines 270-271. "Two of the most important factors for plants' use of N are the availability and demand for N use. The variability of these processes is determined primarily by the BNF and NUE respectively, which are both known to be affected by increased CO2 and N". This statement is not entirely correct. Variability in N demand is not primarily governed by C:N ratio (which is referred to as NUE in the manuscript). C:N ratio of plants changes gradually. The variability in N demand comes primarily from variability in NPP in response to interannual variability in climate. N availability on the other hand depends on pool sizes of ammonia and nitrate. While, BNF is the primary natural mechanism of inorganic input to soil the subtlety here is that pool sizes do not vary substantially from year to year while BNF does. So, I think, variability in N availability has to be very small. Plant N uptake on the other hand will likely be more variable because both passive and active N uptake depend on variability in climate. Please consider rewording this statement.

Page 9, line 275. "The BNF responses to +CO2 of the models differ from the average response recorded in a global meta-analysis of CO2 manipulation (Liang et al., 2016)". Here, Liang et al. is yet another meta-analysis that is being used to evaluate models without properly introducing it first.

Page 9, lines 279-284. This discussion about BNF is hard to follow.

Page 10, lines 300-301. "The large variations in signal and sign of BNF and NUE response between models suggests there is still progress to be made". Perhaps reword this as "The large variations in the magnitude and sign of BNF and NUE responses to +N treatment between models suggests there is considerable uncertainty in our understanding". There are now several meta-analyses (including that of Liang et al. 2016) that clearly show that elevated CO2 leads to increased BNF and studies that show elevated N input decreases BNF. This is also intuitively expected. So, I think, there is sufficient evidence to suggest a real world sign (+ or –) on the response of BNF to these two drivers (+CO2 and +N).

Page 11, line 343, reads "The models mostly represent high latitude northern hemisphere regions less well than other parts of the world, in part because of the unique challenges these areas set for models". I am unsure how can it be concluded that high latitudes are represented "less well than other parts of the world". There are no gridded observations for +N experiment. Does this refer to the fact that the models do not agree at high latitudes. If yes, please say so explicitly.

Page 11, lines 345-349. If I am following the manuscript as the authors intend, it seems the complex processes at high latitudes including potential for release in methane, albedo changes with vegetation expansion, and large amounts of C in soil are mentioned as why the +N response in this region is higher than the average seen in LeBauer and Treseder (2008) meta-analysis. I am not sure if I follow this reasoning because it hasn't been explained how these complex processes are linked to N cycle processes. In addition, were any of the individual studies in the LeBauer and Treseder (2008) meta-analysis performed in the tundra region? If yes, what was their response to +N? What is the northern most study in the LeBauer and Treseder (2008) meta-analysis?

Page 11, lines 357-358 reads "For +CO2 there is the potential for increased NPP because the NUE increases, giving productivity increase without an increase in LAI". I am unable to follow this argument. Isn't is that the productivity increases in the +CO2 experiment simply because of the CO2 fertilization effect? The increase in NPP (due to CO2 fertilization effect) results in a higher C:N ratio of vegetation (which is referred to as NUE), and not caused by C:N ratio as this sentence seems to imply.

Figure 1. Please plot continental boundaries.

Figures 2 and 3. Using similar shades of green and blues for only 5 models is confusing. Please consider using other colours as well.

Figure 3. The arrow for heterotrophic respiration (rh) should come out of the SOM+Litter C pool not the vegetation pool.

Figures 4 and 5. The ratio of small numbers are always misleading and not as meaningful. I am wondering if the geographical plots in Figures 4 and 5 would provide more information if plotted in gC/m2.year rather than percentage change. I realize that the observation-based estimate is in the percentage.

Figure 7. The y-axis titles "BNF response" and "NUE response" are perhaps better written as "BNF change" and "NUE change", although please use C:N ratio instead of NUE .

References other than those that are already in the discussion manuscript

Fageria, N. K. and Baligar, V. C.: Enhancing Nitrogen Use Efficiency in Crop Plants, Adv. Agron., 88, 97–185, doi:10.1016/S0065-2113(05)88004-6, 2005.

---

## Referee Comment (RC2) · P.E.T. Thornton (Referee) · 31 Mar 2020

[referee-annotated manuscript omitted]

---

## Author Comment (AC1) · 17 Jun 2020

The comments of the reviewer are in *italics*, and author responses in blue plain type.

*Authors compare the behaviour of coupled terrestrial N and C cycles in five models that are contributing results to sixth phase of CMIP (CMIP6). The subject of the manuscript is of clear and significant interest to the Earth system modelling community as more and more land components in ESMs explicitly represent terrestrial N cycle and given the large spread among land C cycle models. However, in its current state the manuscript appears to be written hastily with several points unclear, statements that are weakly supported, some incorrect statements, and at places the analysis of results is as simple as which model produces high values of a given quantity and which low.*

This paper was produced under a number of constraints that were sub-optimal, but of course the reviewer should judge the paper on its own merits. We assure the reviewer that this paper is not intended to be disrespectful of reviewer time or the community at large. We aimed to provide a clear appraisal of the models and their performance, without pretentions. These are new models and their collective and comparative performance is not commonly known, and the mechanisms behind the differences are still under investigation. We apologise that the reviewer found some parts difficult to understand and have used the comments to improve the paper's clarity.

*I have three major comments.*

*First, nitrogen used efficiency (NUE) as introduced in equation (1) is simply C:N ratio. In the current literature NUE is typically defined as an efficiency indicator for the utilization of nitrogen in agriculture and food systems (Fageria and Baligar, 2005). That is, higher the NUE the lower amount of applied N enters the environment. I suggest, to avoid confusion with existing definition of NUE authors simply use C:N ratio in equation (1).*

While it is true that in agronomic research NUE is defined as the efficiency of nitrogen recovery per unit added fertiliser, in ecosystem research it is, as in our paper, defined as the growing season integrated nitrogen requirement for growth, or in other words the net primary production per unit nitrogen uptake. While the unit is that of a simply C:N ratio, we note that there is a difference between the NUE and the C:N ratio of vegetation given the different turnover times of the various tissue types, nutrient retranslocation upon senescence as well as plant N inputs due to N fixation. As such, it is a relevant measure of performance of N-enabled terrestrial biosphere models, which has been used by the community. We appreciate this might cause some confusion for non-nitrogen modelling specialists, it seems sensible to follow the established method set for the terrestrial biosphere modelling community rather than switching to a definition used by agronomists.

*Second, the authors have compared the results of two experiments, +CO2 and +N, from models with observation-based estimates. I feel that, the observation-based estimates and the experiments they were based on have not been properly introduced. Nor do the authors discuss limitations of these real world experiments whose results are used to evaluate models. For example, the results from +CO2 experiment used to evaluate models are based on the Baig et al. study which is a meta-analysis but a reader is never told about this. How many studies does this meta-analysis summarizes results from? Similarly, for the +N experiment, the LeBauer and Treseder (2008) study is also a meta-analysis. Both these meta-analyses, results from which are used to evaluate models, should be properly introduced and their limitations discussed. For example, the +CO2 type experiments done are based on instantaneous doubling of CO2 while in the real world CO2 is increasing gradually. Similarly, in +N experiments additional N application rates, I think, are increased instantaneously while in the real world N deposition rates have increased gradually. In addition, can the average*

*results from meta-analysis be used to evaluate the globally-averaged response. The photosynthesis theory says that the CO2 fertilization effect must be strongest in the tropics. How does one account for this? Were the studies used in meta-analysis uniformly distributed geographically speaking? As a modeller myself, I realize, the business of evaluating models is difficult but as long as limitations of observation-based estimates are mentioned, it allows readers (and authors too) to make a rationale and informed expectation of the extent to which observations and models should compare well with each other.*

We understand this concern and have made several changes to ameliorate it. The visual comparisons in (now) figures 2 and 3 have the potential to be misleading, so we have removed those and added a new table (table 3), which contains the same information about the observations but has more room for nuance. We have added a new sub-section to the methods with more details of the observational data used for comparison. We have extended the existing part of the Discussion, which discusses the limitations of the observations, with the reviewer's suggested topics above.

*My third comment is that as a reader, after reading this manuscript, I am not sure if I know anything more about N cycling in models than I did before.*

Since CMIP6 experiments have been so long in being published some well-informed readers, such as the reviewer, will have seen many of the results in conference presentations etc., therefore the results may feel less 'fresh' to them. However, that does not detract from the fact that this information has not been published (in a journal article) before.

The +N and +CO2 experiments are, to our knowledge, the first published results of this kind for a range of LSMs with nitrogen cycles which are used in CMIP6. There is a value in presenting what the state-of-the-art models are doing and comparing them to each other and available metrics, even if we cannot completely explain why the models differ. In the revised version we highlight the implications of fundamental model assumptions regarding spin-up and biological N fixation, which should be informative for future use of the models.

Further, many researchers on the fringes of either CMIP6 or nitrogen in LSMs will find this paper a useful summary of the key results and model features. While senior researchers such as the reviewer may find marginal benefit, there are many more junior researchers who will find it very useful.

*I feel, the results from these models need to be analyzed and reported in a much more clever way to provide overarching conclusions.*

As a modeller, the reviewer will be aware that model comparisons do not always provide the neat generalisations one might hope for. Unlike carbon, nitrogen model structures are very heterogeneous, and their effects are confounded by co-occuring differences in the treatment of the carbon cycle, making simple overarching conclusions inappropriate, as they would be misleading. Therefore, more nuanced conclusions, whilst not "clever", are more scientifically robust. An alternative approach, to implement model assumptions into one common framework (e.g. Meyerholt et al., (2020)), results in a cleaner identification of process importance, but can also rightly be criticised as not representing the effects simulated by the actual CMIP6 model.

*Note that the ability of models to simulate recent trends in GPP and NBP is not due to N cycle. Models without N cycle can achieve this too as is seen in the TRENDY intercomparison which contributes results to annual Global Carbon Project studies (Le Quéré et al., 2018).*

We appreciate the reviewer's sense of humour in pointing out TRENDY simulations when one of the authors of this paper is the lead author for GCP 2019. We included this section because what is notable about these models is that despite incorporating a major new model component to models which simulate GPP/NBP well the models are \*still\* able to simulate recent trends. As the reviewer is no doubt aware, it is not a given that a model will (continue to) perform well after a major change has been made. Examples exist (e.g. Koven et al., (2013)) where the original inclusion of a N-cycle representation limited the ability of the model to reproduce the contemporary carbon balance. However, upon reflection we agree with the reviewer's implied point that this case is made sufficiently in other places already. Therefore, we have moved the first two figures to the SI, removed the associated text, and incorporated brief references to the figures into the main text regarding the N budget.

*Other comments*

*Page 1, abstract, lines 26-28. Upon reading these lines it is clear that 200 ppm CO2 and 50 Kg N/hectare.year N deposition increase are both hypothetical. But as a reader I was wondering what observations are used. At this point in the abstract the reader is not aware that model results are being compared to results from meta-analyses later in the manuscript.*

Obviously the observations used are important, but we do not feel it is appropriate to detail in the abstract all the different sources of observational data. Since meta-analyses are observation-based, we feel this is a fair brief description for the abstract of a paper that focuses on the models.

*Page 3, line 85. Please consider rewording "All models ran a global spin-up for all ecosystem pools up to the year 1860" to "All models pools were spun up to equilibrium using climate and other forcings corresponding to year 1860".*

We have changed this paragraph to make this clearer.

*Page 4, Section 2.2 and 2.3. Please consider summarizing in a table the runs performed. After the pre-industrial spin up, it seems, three runs have been performed – a 1861-2015 historical simulation, a +CO2 simulation for the period 1996-2015, and a +N simulation for the period 1996-2015.*

We have inserted a table of this information to the SI.

*Page 4, equation (1). Please use C:N ratio in this equation as opposed to NUE.*

This equation follows the precedent set in Zaehle et al. (2014a), and we regret that it would not be appropriate to change it and therefore make it inconsistent with previous work on the same topic by (some of) the same authors.

*Page 4, equation (2). This equation is incorrect. Change in NPP cannot be simply determined by multiplying the changes in NUE and N uptake. Please see https://en.wikipedia.org/wiki/Product_rule which explains the product rule of differentiation.*

We apologise for this mistake and have revised this section accordingly and removed this equation.

*Page 4, equation (3). Please define delta N (which implies N balance, I think) properly in words. It seems it is the change in total amount of N in the land (Tg N). But the right hand side terms of the equation are all fluxes which implies the units of N should be Tg N/year. I am confused. The term "N balance" is used throughout the manuscript. It is an important term and yet in the absence of clear worded definition and units it is difficult to follow the context in the rest of the manuscript where this term is used.*

Thank you, we recognise this was not sufficiently clear and have amended the line before equation 3 (now equation 2) appropriately.

*Page 5, lines 136-137 reads "This generation of N models are generally consistent within observational constraints, showing an improvement compared to CMIP5 N models". However, nowhere in the manuscript have model results from CMIP5 models been shown so how can one conclude CMIP6 models are better than CMIP5 models. Please reword this sentence.*

This sentence has been deleted, as it evidently caused more confusion than the benefit of the point which was being made was worth.

*Page 6, line 168. Please consider replacing "non-N model structure" with "C cycle related processes".*

Changed as suggested.

*Page 6, line 174 reads "Across the ensemble there is a slight correlation between the global GPP total and NEP". Please note that for the pre-industrial spin up models' NEP is zero since the model has been spun up to equilibrium. This implies for the pre-industrial state there is no correlation between GPP and NEP. Over the historical period, there is no reason to expect a strong correlation between absolute GPP values and NEP. What is expected is a strong correlation between rate of increase of GPP and NEP since it is the rate at which GPP increases that determines the land C sink.*

*Page 6, lines 175-176 are unclear.*

This paragraph (lines 174- 177) has been removed as it appears to be confusing rather than enhancing the overall message of the paper.

*Page 6, line 186 reads "BNF on the other hand has a wider observed range . . .". For a reader it is unclear where the observed range of BNF comes from.*

Reference added in the text (reference already in Figure 3, referred to in the previous sentence).

*Page 7, lines 202-204 read "Looking at inputs and losses excluding anthropogenic N addition (BNF + N Deposition – N Loss), all the models have a surplus of N and could be said to be 'open' systems with regard to N balance". I am not sure what this means. Recall that after the pre-industrial spin up the sum of all model input N fluxes should ideally be the same as output model fluxes. Was this evaluated? During the transient simulation additional N deposition and fertilizer input leads to increased gaseous losses of N, perhaps increased leaching, and accumulation of N in organic and inorganic pools. I am unsure what 'open' and 'surplus of N' means– does it mean all the additional N input is lost as gaseous fluxes and to leaching. We all know BNF (especially due to increase in crop area) and N deposition increase over the historical period so N balance, as defined in the manuscript, will always be +ve. What's more important here is where does this additional N ends up?*

The issue of whether there is an "open system" with regards to N is a debate popular in some parts of the N community. This comment has made us reconsider whether this is an aspect is worth highlighting and therefore we have removed this sentence rather than enlarge and explain a point that is, perhaps, esoteric.

With regard to the spin-up, as mentioned already in the text of the methods section 2.2, the models were spun-up to equilibrium (as specified by the protocol). The model groups were responsible for ensuring their model was appropriately spun-up and are experienced with running simulations such as these with their models.

*Page 7, line 206 reads " . . . Soil+Litter C is generally low, compared to observational estimates . . .". Does the observation-based estimates contain C from peatlands? The CMIP6 models, I suppose, do not account for C in peatlands and perennially frozen C in permafrost. Could this be the reason for low model estimates.*

*Page 7, line 209. "Comparing the C:N of Soil+Litter global total weight the ratios are similar across models . . .". This sentence doesn't read properly. Also, this section reports the reason for higher C:N ratio of the soil organic matter in the JSBACH model as "The higher ratio for JSBACH is due to the 10:1 ratio for slowly decomposing soil carbon (humus) and larger ratio for litter". I cannot follow what this sentence is trying to imply. This is true for all models. Soil C always decomposes slowly than litter.*

Given the reviewer comments we have given a lot of thought as to which parts of this paper are providing the most pertinent discourse. We decided this paragraph on soil and litter C is not sufficiently useful, given the uncertainties the reviewer mentioned, to warrant it remaining. Therefore, we have removed it.

*Page 7, Section 3.1. In Figure 4, sub panel, it seems the global model response is compared to observation-based estimate from Baig et al. 2015. Is the Baig et al. 2015 average representative of the whole globe or weighted heavily towards certain geographic regions.*

The Baig values are taken from Table 3 in that paper and refers to the % eCa effect on total biomass. It is based on 82 observations of woody plant responses in a variety of locations. The value comes from an equal weighting of each value, regardless of geographical location. We have replaced the reference to Baig et al. (2015) with that from Song et al. (2019) which has more observations and is not limited to woody plants.

*Page 8, lines 226-227 read "Therefore, although the models reach a majority consensus on +CO2 NPP effects overall, the important regional details are still contradictory". Does OVERALL in this sentence means globally? When the manuscript says the "important regional details are still contradictory", I think, it is meant that regional response to +CO2 do not agree amongst models. I think, it doesn't mean they the models contradict some observations because there aren't any regionally aggregated +CO2 observation-based responses. Please reword this sentence.*

Reworded to: "Therefore, although the models reach a majority consensus on +CO2 NPP effects globally, models show contradictory responses for some important regions."

*Page 8, lines 235-239. I wonder, if there a way to quantify or plot this dichotomy between +N and +CO2 responses.*

We have added a new plot and results paragraphs to show this.

*Page 8, lines 255-256. "The largest responses to +N and +CO2 of input and loss do not necessarily correlate with either N uptake or changes to productivity". I am not sure what this sentence means.*

We have revised and hopefully clarified this sentence and the rest of the paragraph.

*Page 9, line 267. "In contrast, JSBACH has less than half the increase in loss of JULES in the +N simulation". By the time, a reader reaches this sentences he/she may forget what quantity is being referred to. Does this sentence refers to plant N uptake?*

We hope that the clarification of "N loss" rather than "loss" will help aid the reviewer's understanding. The entire context helps, as the previous sentence discusses N loss, and since the

sentence begins, "in contrast", we hope it will now be clear: "In common with all the models, in JULES the N loss term is a fixed fraction of the mineralisation flux and the soil N pool size. In contrast, JSBACH has less than half the increase in N loss of JULES in the +N simulation (Fig. 6c) and almost no change in NUE (Fig. 7d)."

*Page 9, lines 270-271. "Two of the most important factors for plants' use of N are the availability and demand for N use. The variability of these processes is determined primarily by the BNF and NUE respectively, which are both known to be affected by increased CO2 and N". This statement is not entirely correct. Variability in N demand is not primarily governed by C:N ratio (which is referred to as NUE in the manuscript). C:N ratio of plants changes gradually. The variability in N demand comes primarily from variability in NPP in response to interannual variability in climate. N availability on the other hand depends on pool sizes of ammonia and nitrate. While, BNF is the primary natural mechanism of inorganic input to soil the subtlety here is that pool sizes do not vary substantially from year to year while BNF does. So, I think, variability in N availability has to be very small. Plant N uptake on the other hand will likely be more variable because both passive and active N uptake depend on variability in climate. Please consider rewording this statement.*

We have revised these sentences and much of the rest of the paragraph to enhance clarity.

*Page 9, line 275. "The BNF responses to +CO2 of the models differ from the average response recorded in a global meta-analysis of CO2 manipulation (Liang et al., 2016)". Here, Liang et al. is yet another meta-analysis that is being used to evaluate models without properly introducing it first.*

As mentioned above, we have introduced a new methods sub-section in the methods to introduce all the observations used as comparisons.

*Page 9, lines 279-284. This discussion about BNF is hard to follow.*

We apologise that this section was hard to follow and have revised the text thoroughly.

*Page 10, lines 300-301. "The large variations in signal and sign of BNF and NUE response between models suggests there is still progress to be made". Perhaps reword this as "The large variations in the magnitude and sign of BNF and NUE responses to +N treatment between models suggests there is considerable uncertainty in our understanding". There are now several meta-analyses (including that of Liang et al. 2016) that clearly show that elevated CO2 leads to increased BNF and studies that show elevated N input decreases BNF. This is also intuitively expected. So, I think, there is sufficient evidence to suggest a real world sign (+ or –) on the response of BNF to these two drivers (+CO2 and +N).*

Text adjusted as suggested.

*Page 11, line 343, reads "The models mostly represent high latitude northern hemisphere regions less well than other parts of the world, in part because of the unique challenges these areas set for models". I am unsure how can it be concluded that high latitudes are represented "less well than other parts of the world".*

The evidence is given in the results section and this sentence has be reworded to refer the reader back to the appropriate section.

*There are no gridded observations for +N experiment. Does this refer to the fact that the models do not agree at high latitudes. If yes, please say so explicitly.*

While there are no gridded observations for +N, there are (as shown in Fig. 6) three separate estimates for +N for Tropical, Temperate, and Boreal regions. We concede that those biomes do not cover the whole globe, but think it is a fair statement. It is an attempt to draw a tentative "overarching conclusion" of which region is most challenging for all N models. The reviewer mentions in their first points about this paper that they would like more "overarching conclusions" so presumably would support this.

*Page 11, lines 345-349. If I am following the manuscript as the authors intend, it seems the complex processes at high latitudes including potential for release in methane, albedo changes with vegetation expansion, and large amounts of C in soil are mentioned as why the +N response in this region is higher than the average seen in LeBauer and Treseder (2008) meta-analysis. I am not sure if I follow this reasoning because it hasn't been explained how these complex processes are linked to N cycle processes.*

We apologise that this point was not as clear as it ought to have been, as the reviewer has misunderstood. We have rephrased this paragraph and hope the point is now clearer.

*In addition, were any of the individual studies in the LeBauer and Treseder (2008) meta-analysis performed in the tundra region?*

Yes, there is n=10 for Tundra in the LeBauer and Treseder (2008) study (see Table 1 in that paper).

*If yes, what was their response to +N?*

35% increase in aboveground net primary productivity (LeBauer and Treseder (2008), Table 1).

*What is the northern most study in the LeBauer and Treseder (2008) metaanalysis?*

78˚North, a Tundra site on Ellesmere Island (LeBauer and Treseder (2008), Fig. 1).

*Page 11, lines 357-358 reads "For +CO2 there is the potential for increased NPP because the NUE increases, giving productivity increase without an increase in LAI". I am unable to follow this argument. Isn't is that the productivity increases in the +CO2 experiment simply because of the CO2 fertilization effect? The increase in NPP (due to CO2 fertilization effect) results in a higher C:N ratio of vegetation (which is referred to as NUE), and not caused by C:N ratio as this sentence seems to imply.*

We have reworded this sentence to make this point clearer.

*Figure 1. Please plot continental boundaries.*

We have added continental boundaries to the map figures.

*Figures 2 and 3. Using similar shades of green and blues for only 5 models is confusing. Please consider using other colours as well.*

We are sorry the reviewer found the viridis colour palate (yellow, light green, dark green, blue, and purple) confusing. Viridis was designed to be comprehensible to a wide range of visual impairments and also converts well to grey scale to allow the paper to be printed black and white to save costs and environmental impact. See https://cran.r-project.org/web/packages/viridis/vignettes/intro-to-viridis.html for further information.

*Figure 3. The arrow for heterotrophic respiration (rh) should come out of the SOM+Litter C pool not the vegetation pool.*

This has now been corrected, thank you for noticing this mistake.

*Figures 4 and 5. The ratio of small numbers are always misleading and not as meaningful. I am wondering if the geographical plots in Figures 4 and 5 would provide more information if plotted in gC/m2.year rather than percentage change. I realize that the observation-based estimate is in the percentage.*

We have added the figures showing absolute amounts for the +N and +CO2 experiments as latitudinal averages to the SI and referred the reader to these plots in the main text. While we agree that the absolute numbers are useful, given that most observational comparisons are a percent change, this is most useful for the main figures.

*Figure 7. The y-axis titles "BNF response" and "NUE response" are perhaps better written as "BNF change" and "NUE change", although please use C:N ratio instead of NUE .*

"Response" has been changed to "change" in Fig. 7 (now Fig. 5). For consistency, as discussed above, we would prefer to leave NUE.

*References other than those that are already in the discussion manuscript Fageria, N. K. and Baligar, V. C.: Enhancing Nitrogen Use Efficiency in Crop Plants, Adv. Agron., 88, 97–185, doi:10.1016/S0065-2113(05)88004-6, 2005. Interactive comment on Biogeosciences Discuss., https://doi.org/10.5194/bg-2019-513, 2020.*

References mentioned in response:

Koven, C. D., Riley, W. J., Subin, Z. M., Tang, J. Y., Torn, M. S., Collins, W. D., Bonan, G. B., Lawrence, D. M. and Swenson, S. C.: The effect of vertically resolved soil biogeochemistry and alternate soil C and N models on C dynamics of CLM4, Biogeosciences, 10(11), 7109–7131, doi:https://doi.org/10.5194/bg-10-7109-2013, 2013.

Meyerholt, J., Sickel, K. and Zaehle, S.: Ensemble projections elucidate effects of uncertainty in terrestrial nitrogen limitation on future carbon uptake, Global Change Biology, n/a(n/a), doi:10.1111/gcb.15114, 2020.

---

## Author Comment (AC2) · 17 Jun 2020

The comments of the reviewer are in *italics*, and author responses in blue plain type.

*The topic of the paper is relevant to the ongoing evaluation of the sign, magnitude, spatial variability, and potential future trajectory of land ecosystem feedbacks among increasing CO2, increasing N deposition, physical climate variables, and the global scale cycling of carbon and nutrients.*
*Some results presented here could help to inform the evaluation of existing models or the development of new modeling approaches, in particular the results summarized in Figure 8.*
*On the whole, the results are presented in the form of assertions without adequate quantitative support, and without sufficient process-level elucidation of either the behavior of individual models, the differences between models, or the relationship between models and observation-based datasets. The manuscript overall suffers for having too much description and not enough explanation.*
We aimed to provide an honest appraisal of the models and their performance, without affectations. These are new models and their collective and comparative performance is not commonly known, and the mechanisms behind the differences are still under investigation. We have comprehensively reworked the paper with additional analysis and added quantitative references in convenient locations for the reader.

*L53. Either "an important source", or "the main source".*
Changed to "an important source".

*L54. Meaning not clear*
We have changed the phrasing to "overall performance" which we hope will be easier to comprehend.

*L56. CLM?*
We presume the reviewer means to point out that CLM is not European, and we absolutely did not mean to imply that. We were careful in our phrasing of "European ESMs" (not LSMs). We have clarified this to "of ESMs used in European Earth System modelling centres". We further clarify this in section 2.1: CLM4.5 is used within CMCC-CM2, and CLM5 is used within NorESM. The reason for specifying European ESMs is to give some rationale for the choice of LSMs. This project is centred on the EU Horizon 2020 project CRESCENDO so this is useful in leading the reader to the reason why other models are not included.

*L69. Add citations to Thornton et al. 2007, 2009*
Citations added.

*L74 Does not appear in bibliography.*
In the version reviewed this reference is at line 613 of the reference list (there is no bibliography). In the revised version this reference can be found between Sellar and Smith.

*L87 land cover*
Changed.

*L96 Can you use the current results to test that assumption?*
Unfortunately not, as that would require a whole new set of simulations (with identical initial states) that we do not have. The model protocol was carefully set up and was designed to test hypotheses

about nitrogen parts of the model and we believe it to be robust. While the question of this assumption could be interesting, it is beyond the scope of this particular project.

*L116 Does this approach account for retranslocated N that is taken up once, then stored in the plant and reused for multiple growing seasons?*
Note that this is a diagnostic calculation and is based on the modelled rates of NPP and Nuptake. Therefore, it does implicitly account for retranslocated N as dealt with by the individual models, as the rates of retranslocated N in year x affects N uptake in year x+1. Retranslocated N is considered to be within the plant and thus utilised (even if that utilisation is as labile N stores). Incorporating retranslocation into analysis would be challenging, as the models deal with retranslocation in a variety of ways that are not easily comparable.

*L123 The meaning here is unclear. If allocation patterns at the plant level shift between tissues with different C:N, wouldn't NUE also shift?*
We have revised this section to aid comprehension.

*L153 There are many different kinds of model deficiencies that could lead to such mismatches. There are also potential deficiencies in the data that could explain the differences you observe. What can you say, mechanistically, about the models or about the data that might shed some light on the possibilities?*
*L156 A couple of issues with this paragraph. First, looking at the Fig 1a and 1b results, the differences between the two models outside the high latitudes seem even more pronounced - for example over the African tropics, Indonesia, southeast Asia, and the southeast of North America. Second, this paragraph simply lists a range of differences between the models, but doesn't do anything to try to explain why those differences might be related to differences in GPP in particular regions, or how any of that might relate to differences in connections between nitrogen cycling and GPP.*
*L165 I don't understand why the model results in Fig 2 are normalized as anomalies with respect to their own 1901-1910 period. The GCP assessments shown in black in Fig 2 do not include that same kind of anomaly calculation. The models should be showing losses of carbon from land during the early stages of the simulations, due to the effects of land use and land cover change. So they are probably all at rather different values of NEP in the 1901-1910 period (although hopefully these raw values would mostly be on the negative side). In other words, It is the raw NEP that should be compared to the GCP numbers.*
*L168 This paragraph is mainly composed of unsupported assertions. Yes, the two CLM variants are the lowest of the five models for the (anomaly-adjusted) NEP. See above comment for why the anomaly adjustment makes even this apparently simple assertion difficult to assess. The assertion that differences in GPP are larger than differences in NEP, and that spatial patterns should be considered, is not supported by any quantitative assessment, and since there is no spatial map of NEP it's not possible to assess that part of the comparison. Finally, the assertion that the supposed similarity in NEP in contrast to differences in GPP are caused by similar N effects on the respiration terms is pure speculation, with no supporting analysis.*
*L175 This paragraph summarizes a few rank relationships, but offers no mechanistic explanation, other than an unsupported claim that considering more processes increases uncertainty.*
Upon reflection of the comments of reviewer 1 we have removed this section and included the figure on GPP in the SI, as the stronger part of the paper is the +N and +CO2 experiments and that ought to be the focus. The reviewer is correct that the model NEP ought not to be normalised to 1901-1910. This part of the paper was written before the publication of the latest GCB paper (Friedlingstein et al. 2019), but since the publication of the final version of this paper we have decided that this figure is defunct and have removed it altogether.

*L183 Consistency with what?*
Clarified as "the internal consistency of this schematic".

*L190 One useful contribution could be to try to quantify these differences, by showing a few details of the implementation, together with differences indriving variables (NPP or ET) among models.*
A "few details of the implementation" of the BNF are available in Table 1. The driving variables (NPP or ET) are only relevant to three of the five models and GPP is already available (in what was figure 1 and is now in the Supplementary Information). The papers by Wieder et al. (2015) and Meyerholt et al. (2016) that we cite already offer the useful contribution of quantifying the differences between different BNF representations.

*L197 So, are you suggesting that the comparison to observed values would be even worse if the observed values had not been corrected for root respiration? I'm not clear on the meaning of "however" in this sentence.*
We appreciate the reviewer drawing our attention to this not being phrased as clearly as it ought to have been and have adjusted the text accordingly.

*L204 It isn't clear what the significane of being an "open" system is. The models are accumulating carbon on land in the period summarized in Fig 3. They are also accumulating N.*
This point was not clear to the other reviewer either, so in the interests of focusing on the most important areas of the work we have removed this point.

*L209 This comparison could be more informative if there was a table of C:N by model and pool, and the accompanying C totals py model and pool.*
We have added the C:N ratios to figure 1 (previously figure 3).

*L211 It is a stretch to make this statement based on the analyses presented. There are lots of additional sources of observations for both fluxes and pools.*
This sentence has been removed.

*L218 The cited study looked at woody species only. Are the model results being weighted somehow in this analysis to emphasize the influence of woody biomes?*
We have replaced the comparison with Baig et al. (2015) with a comparison with Song et al. (2019). The latter is not vegetation type specific.

*L226 It would be useful and interesting to examine why these differences occur. A mechanistic investigation of the differences in model structure or parameterization. Some relationship to nutrient cycling, perhaps?*
At this point it is not clear why this occurs, and we have added a statement to that effect to the text.

*L230 It could be interesting to contrast these models in terms of their development of N limitation over time. That would mean including some time series outputs.*
This would indeed be very interesting, but we note that N limitation is an emergent outcome of the model simulation and not readily quantified by any modelled quantify that can be easily compared across models. Using reference simulations with a C-cycle only representative would also be a plausible mean for any one given model, however, also here the across model comparison would not be straight forward.

*L238 Not clear what this means, or why it is significant. The spatial patterns for +CO2 and +N for JSBACH seem quite different. Is that a dichotomy?*
We have added a new figure and results section that develops this point. You will see from that figure that while the effect in JSBACH is less pronounced, the homogeneous distribution of +CO2 and +N results that "quite different" spatial patterns would give, is absent.

*L243 It would be better to have a table where the results by model are summarized for different biomes. Then the last sentence of this paragraph would have some quantitative basis. As it is, several of the models appear to have high +N effects in grassland regions, but that is just my crude analysis "by eye" of the results shown in Fig 5. Surely you can do better, having access to all the model outputs.*
A table of these values, by biome, has now been included in the SI.

*L253 I am surprised that such a cursory sort of analysis is included as a result. If you want to assert a correlation, even a weak one, go ahead and plot the data or show us a regression result. I think you would find that the correlation is very weak indeed, in this case, if you made that effort.*
This sentence has been removed.

*L257 This statement is confusing. Looking at Fig 6, it looks to me like the CLM4.5 response for BNF and N balance are the smallest of all models, not middle of the range.*
Apologies, this has been corrected to gaseous loss and leaching and uptake.

*L260 So are you saying that this is just a difference in accounting? If so, can't that be corrected for in the analysis to make a more useful comparison?*
Done.

*L264 This is a good example of a process-based description of the differences between models.*
We are glad you appreciate this.

*L270 The assertions that N availability is primarily controlled by BNF, and that N demand is primarily driven by NUE are unsubstantiated here. One could just as easily say that availability is controlled more by mineralization rate, or by low levels of loss - without some quantitative assessment to back this up it is just a conjecture. Similarly, why not say that N demand is driven mainly by GPP? Why tie the causality statement for N demand to a term that has N uptake as its denominator?*
This paragraph was intended to be a helpful introduction to the section on BNF and NUE, giving an overview of why we focus on those two metrics, without overburdening the reader with extensive background, references, and explanations as why these two metrics are the focus. Since this short paragraph is evidently causing confusion rather than smoothing the transition of topic, we have opted to remove it.

*L285 It would be helpful to say something here about the observations: how representative do you expect they would be of a true global mean response?*
There is a new section in the methods detailing the observations used and extensive discussion of the robustness (or not) of the observations in the discussion section. While we understand that the uncertainties about the observations are important, we prefer for the results section to primarily contain the results of our work.

*L287 Saying it is counter-intuitive is different than saying it is counter-factual. Here is an example: in a shrub tundra community dominated by alder (an N fixer), there may still be significant N limitation. By*

*fertilizing the site, it is possible that the alder could expand its occupancy, with an overall increase in NPP. Assuming a constant NUE (for the sake of argument), a higher NPP would mean a higher N demand. It could well be that with higher N input and higher N demand, the community might still experience significant N limitation. In that case increase of BNF with increase in NPP under increased N input seems a plausible outcome.*

The phrase "counter-intuitive" has been removed and the sentence rephrased. As the reviewer rightly says, there are a small number of situations where fixation might continue in the presence of increased N supply. The reviewer gives the example of alder, which is an obligate N fixer. However, N fixers can also be facultative, and thus able to slow or stop BNF where increased N supply makes it not energetically worthwhile (Menge et al., 2009). In the figure caption we cite observations (Zheng et al., 2019) at the larger biome scale, that show BNF reduces with increased N deposition. We have added a citation to the main text so the reader can more easily see the evidence that it is "counter-factual" that at the macro scale models increase BNF with increased N supply.

*L291 It is in line the current theory of Walker et al., 2015.*
Changed as suggested.

*L293 Again, couldn't you address this in the analysis to make for a more internally consistent comparison?*
We have added BNF to Nup for CLM5 as suggested. In the process of amending the code we found a small mistake in the JULES-ES results which we have now corrected. This has prompted us to review all the other figures and we found no further issues.

*L299 It would be better to state this as a hypothesis, and provide an explanation for why you think this is the likely behavior of real ecosystems.*
As suggested, we have rephrased this as a hypothesis and enlarged on this point.

*L302 These results are your most interesting material.*
Thank you for this considerate comment. Upon reflection we agree and have increased this section correspondingly and removed some other extraneous material.

*L330 Has anyone tried to disprove this? It seems like the kind of model-generated hypothesis that would be testable in laboratory conditions using pre-industrial CO2 concentrations and N fertilization. Some actual discussion of this would be interesting.*
So far as we know this is an ongoing debate in the N modelling community. We have added to the discussion of this point by providing each side of the discussion.

*L345 You should at least try to connect the sign of the supposed bias (overestimation of NPP response to +N) to the factors you've listed. Otherwise, you seem to be saying that the response is overestimated because the system is complex. Why not underestimated?*
We have rephrased this paragraph to enhance ease of comprehension.

*L356 Is the NUE increase coming from shifting allocation, or shifting C:N? and is this a statement of what the models are doing, or how the real ecosystems respond? Couldn't +N also cause an increase in production without increasing LAI, for example through more fine root allocation?*
This sentence has been rephrased to aid comprehension.

*L381 I agree.*

We're glad this statement meets with your approval.

References

Menge, D. N. L., Levin, S. A. and Hedin, L. O.: Facultative versus Obligate Nitrogen Fixation Strategies and Their Ecosystem Consequences., The American Naturalist, 174(4), 465–477, doi:10.1086/605377, 2009.

Zheng, M., Zhou, Z., Luo, Y., Zhao, P. and Mo, J.: Global pattern and controls of biological nitrogen fixation under nutrient enrichment: A meta-analysis, Global Change Biology, 25(9), 3018–3030, doi:10.1111/gcb.14705, 2019.

---

## Referee Report (RR1)

I get the awkward position of being a new reviewer introduced to a paper mid-review. Like being a step-parent trying to balance my way of doing things with the fact that the kids (authors) have already developed in another system.

Overall, this is a great paper. It is very challenging to take on a paper that not only deals with a lot of complexities and nuances within the models and observations, but also the fact that one can present the analyses/results in a gazillion different ways, making it hard for readers to absorb. The authors did an excellent job of distilling analyses, results, and interpretations, which make this paper a valuable contribution to the literature.

The biggest challenge is probably benchmarking N cycle impacts against a lot of C cycle measurements. Moreover, the authors do a lot of comparing model outputs to observational ranges; but, we know very well (and the authors discuss briefly in the Discussion), these magnitudes change with choice of forcing data (and other model run conditions). So, then how useful is it to make these direct comparisons? Is there not a different/better way of doing these evaluations that accounts for the fact that the end number changes so easily? The sensitivities and directions should mostly be the same no matter what forcing. I don't expect the authors to change their results at this point out of sheer exhaustion/frustration/workload related to this comment. Still, hopefully a next paper can consider this comment to advance the types of analyses done. That said, the evaluations/analyses done in this paper are much better than what is often done in other papers (e.g., let's just compare to LAI and say the difference is due to the one component that I developed in the model…).

- Abstract
  - Somewhere say that you ran the models offline with common spin-up and forcing protocols—this is very valuable for understanding model differences.
  - L26-28. Maybe put something quantitative to complement the qualitative sentences, something readers can grab as take-home stats.
  - L29-31. It would be amazing to add why…
  - L31. "better represented" is vague/unclear.
  - L33. Throw away sentence. Delete.
  - L34. "better understanding and more provision" is vague/should be more explicit.

- Introduction
  - L41. "allowable" is that the right word? More like nothing or everything is allowed. Projections are just whatever scenarios ESMs are presented with.
  - L59-61. Break up this long sentence?
  - L60. It doesn't totally make sense why this study is limited to European centres. You commented on that in response to one of the reviewers, but it doesn't make sense in the paper. The abstract/title and everything else up to this point seems like the paper is generalizable across the global modeling community. But, then this gets inserted that throws the direction off with a jolt.
  - L62-63. Cite.

- Methods
  - L87. Any update to Wiltshire et al forthcoming? How about a conference abstract?

- Results
  - Fig 1. Cool figure. I wonder if there is more room for artistry in it so that one can visualize the numbers and spread without having to do the math in one's head individually for each component. Could be quite powerful if you can figure it out (it's already quite powerful though, so don't get me wrong).
    - The arrows for higher/lower than obs are nice. BUT, when you have no arrow it means either that it's within range, or that there are no obs. So, you've got some confusion there in symbology.
    - Is there no uncertainty on Ndep obs?
    - It's weird that Ndep differs between models, when they were all forced with the same amount. I guess you explain it with differences in land fraction, but it's still weird.
    - What about having all the obs be a number plus/minus a number. Instead of having some be ranges. Or vice versa.
    - The yellow is hard/impossible to read. Pretty much leaves me "guessing" on those numbers… (sorry, just fishing for a comment on my humor, given that you were giving out those compliments to other reviewers…).
    - Why no model numbers for Nmin, Nup, and Soil Ninorganic?
  - L164. Perhaps a slight bit more elaboration on CLM5's BNF could be useful, as it does seem to be quite different than the other 4 models.
    - Maybe include discussion of CLM4.5 here too, given that you discussed all the models but CLM4.5?
  - L184. Guess→GUESS. Actually, there's inconsistency on this throughout the paper, so just do a find and replace and pick one.
  - Section 3.2-3. Are the Song et al numbers comparable in terms of global scale, temporal scale, $CO_2$, and climate? It seems from Song et al's Fig 1, the data are mostly geographically not where the models are being impacted most at the global scale (e.g., low for JSBACH/JULES-ES, or high for CLM's, LPJ-GUESS). If they're not comparable, then don't compare them. Throw Song et al in to the Discussion or something saying about what would be needed to make them useful.
  - Figure 4. Maybe put somewhere on the figure that we're looking at NPP (in addition to the caption)? Would be good to have this figure stand alone.
    - Maybe make the dots bigger? E.g., it's hard to see JSBACH and JULES-ES.
    - Is this plus/minus latitude? Or just N. Hemisphere? If it's plus/minus, then that really isn't clear in the figure.
  - Figure 5. The red/purple areas are hard to distinguish from one another. Same goes with the orange/yellow, though that's easier as they're more distinct geometrically.
    - Why is there no left purple solid line?

- I'd consider ditching the dashed line altogether. It's really just extra information that isn't even used because the models mostly get nowhere even near the bar areas. The reader can assume the middle point.
  - Figure 6. Cool figure. I'm confused in b and c though. They appear to be showing the N response. But, the text in L262-271 refers to the NPP response.
  - Figure 7. You introduce Fig 7d first, then 7b, and never 7a.
    - Not sure if the publishing editors will pick this up, but sometimes you have a period after Fig, other times not.
    - L280-282. I'm not following this text as it relates to the Figure. The text refers to Fig 7b. It says that JULES-ES is within range of the obs (except boreal). When I look at 7b, I don't see JULES-ES's bar inside the gray bars. Am I interpreting this incorrectly? Same goes with the statements on CLM5. You say that it's a clear outlier with a large increase in BNF. But, 7b shows a large decrease, plus it's kind of similar to LPJ-GUESS. There is an increase in 7a, but one could also just say that all the models are outliers relative to the obs, *except* for CLM5 in the boreal, which it actually hits.
    - I know CLM5 best mostly because I know FUN. So, this is a question specifically from J. Fisher to R. Fisher: how much of the CLM5 N response is due to issues with CLM's C-cycle, i.e., too much GPP/NSC/not enough Rh? It's great to see that CLM5 is going in the right directions etc., but it also looks like the N cycle is hyped up on sugar, like a kid on Halloween. If you cut that GPP down, then you have less C to pay for BNF etc.
  - L300-302. Grammar edit.

Good work overall! I hope my comments are useful.

Josh Fisher

---

## Referee Report (RR2)

Re-review of bg-2019-513
Nitrogen Cycling in CMIP6 Land Surface Models: Progress and Limitations

I thank the authors for making their manuscript easier to follow. The manuscript is in much better form but I am afraid there are still some errors and seemingly incorrect misinterpretations of the data that need to be addressed before it can be accepted for publication. These are mostly minor. I also have several suggestions (some of which are personal choices on how a sentence may be phrased) that I have marked on the manuscript itself whose scanned version is attached.

Major comments

1. Please report the key numbers in the abstract including that the average +CO2 response of the models is X% compared to observations (Y%), and similarly for +N response.

2. One key analysis that is missing seems to be the comparison of late 20$^{th}$ century sink. It should be pretty straightforward to compare the time series of net atmosphere-land CO2 flux from the five models with estimates from the latest Global Carbon Project (GCP) numbers for the decades of 1960s, 70s, 80s, 90s, and 2000s (https://essd.copernicus.org/articles/11/1783/2019/, their table 5). The range and average sink over the period 1960-2010, from the model, can also be reported in the abstract (since this is also a key number) and compared with the GCP's estimate.

3. Right now a large fraction of the Conclusions section seems like part of the Discussion since it discusses the performance of the individual models just like in the Discussion section. I think, it would be helpful if the Conclusion section is more generalized.

4. On page 9, lines 279-289. These lines discuss Figure 7a (+CO2 response) but the text (line 279) says they discuss Figure 7b (the +N response).

5. Page 10, lines 300-303 read "Since the BNF in JULES is directly related to NPP, so the reduction in NUE indicates excess N in the system from mineralisation, possibly related to soil warming, in boreal regions with +CO2, leading to decreased N uptake." This sentence attempts to explain the decrease in NUE of the JULES model for the +CO2 scenario in Figure 7c. This appears to be an incorrect explanation since, I am wondering, how can the soils warm in this offline experiment which is driven with specified meteorological data, compared to the Control run. It seems there has to be some other explanation.

Minor comments

6. The colour scheme for the five models can be better. I find it hard to differentiate between CLM 4.5 and CLM5, and CLM5 and JSBACH. Also, the yellow colour LPJ-GUESS is not readable at all on the grey boxes in Figure 1.

7. Figure 2 in SI. Do the results show model minus observations, or observations minus model?

8. In context of comparing observations to model results, the manuscript doesn't explain what does "upscaling" of observations means and how it is done.

9.  The phrase "dynamic vegetation" (e.g. on line 343), I think, is meant to imply competition between different plant functional (or vegetation) types. If yes, say this explicitly since prognostic LAI, for example, is also an example of vegetation dynamics.

10. Page 4, lines 118-120 read "the net ecosystem balance of N, which determines the change in the N capital available for plant growth and soil organic matter decay". This sentence doesn't read properly and, I think, is incorrect in saying that N balance is the "N capital available for plant growth". Clearly, we know that net N balance is given by

$$\Delta N = N_{dep} + BNF + N_{loss} = \Delta N_{veg} + \Delta N_{soil+litter} + \Delta N_{mineral\_pools}$$

so not all of the N balance is the "capital available for plant growth".

11. Equation (1) in the manuscript, and the analysis in the paper, discusses the inputs ($N_{dep}$, $BNF$) and outputs ($N_{loss}$) but not the changes in pool sizes ($\Delta N_{veg}, \Delta N_{soil+litter}, \Delta N_{mineral\_pools}$). This would have been helpful in investigating how the N balance is split across the organic and inorganic pools in different models but the paper is okay without these too.

12. Page 11, lines 342 and 343 read "direct control of NPP by N availability, whereas photosynthetic C uptake (GPP) is not directly affected by N" in context of JULES and JSBACH showing little productivity response to increased N availability. Since NPP = GPP – Ra, I am struggling to figure, how can NPP be controlled by N availability but not GPP.  The only way this can happen is Ra is controlled by N (through N content of vegetation components) in which case can this be made more clear.

**Nitrogen Cycling in CMIP6 Land Surface Models: Progress and Limitations**

Taraka Davies-Barnard[1,2], Johannes Meyerholt[2], Sönke Zaehle[2], Pierre Friedlingstein[1,3], Victor Brovkin[4], Yuanchao Fan[5,6], Rosie A. Fisher[7,8], Chris D. Jones[9], Hanna Lee[5], Daniele Peano[10], Benjamin Smith[11,12], David Wårlind[11,12], and Andy Wiltshire[9]

[1]University of Exeter, Exeter, UK
[2]Max Planck Institute for Biogeochemistry, Jena, Germany
[3]Laboratoire de Meteorologie Dynamique, Institut Pierre-Simon Laplace, CNRS-ENS-UPMC-X, Departement de Geosciences, Ecole Normale Superieure, 24 rue Lhomond, 75005 Paris, France
[4]Max Planck Institute for Meteorology, Hamburg, Germany
[5]NORCE Norwegian Research Centre, Bjerknes Centre for Climate Research, Bergen, Norway
[6]Harvard University, Cambridge, USA
[7]National Center for Atmospheric Research, Boulder, Colorado, USA
[8]Centre Européen de Recherche et de Formation Avancée en Calcul Scientifique, Toulouse, France
[9]Met Office Hadley Centre, Exeter, UK
[10]Fondazione Centro euro-Mediterraneo sui Cambiamenti Climatici, Bologna, Italy
[11]Department of Physical Geography and Ecosystem Science, Lund University, Lund, Sweden
[12]Hawkesbury Institute for the Environment, Western Sydney University, Richmond, Australia

Correspondence to: T. Davies-Barnard (t.davies-barnard@exeter.ac.uk)

**Abstract.** The nitrogen cycle and its effect on carbon uptake in the terrestrial biosphere is a recent progression in earth system models. As with any new component of a model, it is important to understand the behaviour, strengths, and limitations of the various process representations. Here we assess and compare five models with nitrogen cycles that are used as the terrestrial components of some of the earth system models in CMIP6. We use a historical control simulation and two perturbations to assess the models' nitrogen-related performance: a simulation with atmospheric carbon dioxide 200 ppm higher, and one with nitrogen deposition increased by 50 kg N ha$^{-1}$ yr$^{-1}$. There is generally greater variability in productivity response across models to increased nitrogen than to carbon dioxide. Compared to observations, two models of the models considered here have low productivity response to nitrogen, and another one a low response to elevated atmospheric carbon dioxide. In all five models individual grid cells tend toward bimodality, with either a strong response to increased nitrogen or atmospheric carbon dioxide, but rarely to both to an equal extent. However, this local effect does not scale to either the regional or global level. The global and tropical responses are generally better represented than boreal, tundra, or other high latitude areas. These results are due to divergent though valid choices in the representation of key nitrogen cycle processes. They show the need for better understanding and more provision of observational constraints of nitrogen processes, especially nitrogen-use efficiency and biological nitrogen fixation.

*[Handwritten annotations:]*
- *than WHAT?*
- *compared to what?*
- *to exhibit*
- *though valid choices — availability*
- *compared to observations.*
- *It is hard to say if model processes are valid without reviewing them in detail.*

*Of the ESMs that contributed results to CMIP5 only two ─ ─ ─ ─ ─*

**1 Introduction**

The terrestrial carbon (C) cycle currently removes around a third of anthropogenic carbon emissions from the atmosphere (Friedlingstein et al., 2019; Le Quéré et al., 2018). Changes in this uptake will affect the allowable emissions for targets such as limiting warming to 1.5°C (Millar et al., 2017; Müller et al., 2016). Nitrogen (N) is required to synthesise new plant tissue (biomass) out of plant-assimilated C, in differing ratios across biomes and tissue types (McGroddy et al., 2004). Therefore, future projections of terrestrial C uptake and allowable emissions are dependent on N availability, particularly under high atmospheric carbon dioxide ($CO_2$) conditions (Arora et al., 2019; Meyerholt et al., 2020; Wieder et al., 2015b; Zaehle et al., 2014b). A key tool for projections of allowable emissions are Earth System Models (ESMs), which project the responses of the coupled earth system *to perturbations in climate forcings* (Anav et al., 2013; Arora et al., 2013; Friedlingstein et al., 2006; Jones et al., 2013). The Fifth *reword*

Phase of the Coupled Model Intercomparison Project (CMIP5, Taylor et al., 2012) had numerous ESMs with a global C cycle but only two, based on the same land component, with terrestrial N cycling (Thornton et al., 2009). A number of studies with stand-alone terrestrial biosphere models (Sokolov et al., 2008; Wårlind et al., 2014; Zaehle et al., 2010; Zhang et al., 2013) as well as post-hoc assessments of CMIP5 projections suggest that predictions of terrestrial C storage would *uptake?* decrease by 37 – 58% if ESMs accounted for N constraints (Wieder et al., 2015b; Zaehle et al., 2014b). *terrestrial*

*Among* 50 The latest generation of models *that contributed results* in CMIP6 (Eyring et al., 2016) *to* 
[revised manuscript text omitted]

$$\text{Units of NUE} = \frac{gc\ m^{-2}\ day^{-1}}{gN\ m^{-2}\ day^{-1}} = \frac{gc}{gN}$$

Whether and how this change in N capital affects plant growth is dependent on the magnitude of the change in plant N uptake, as well as relationship between $N_{up}$ and NPP (whole-plant nitrogen-use efficiency; NUE; (Zaehle et al., 2014a))

$$NUE = \frac{NPP}{N_{up}}$$  (2)

where $N_{up}$ includes plant uptake of soil inorganic N of any origin, i.e. atmospheric deposition, fertilization, decomposition of plant litter, or biological nitrogen fixation (BNF). NUE is the outcome of the product of tissue stoichiometry and fractional allocation of NPP to different tissue types, and therefore varies with changes in the allocation fractions and tissue C:N.

**2.5 Observations for Comparison**
*We compare the models against*

a range of observation-based metrics  *the* at global and regional scales, detailed in Table 2. Most of these are based on small numbers of field studies upscaled or averaged to give an approximate global value with confidence intervals. While these upscaled values need to be interpreted with proportional caution, in the absence of more robust comparators they are useful benchmarks that can provide real-world context in addition to field scale comparisons and inter-model comparisons. *What does 'upscaling' means? Who did it?*

### 3 Results

**3.1 Control Run Global C and N budgets**

*at the global scale*

A range of pools and fluxes from the models compared to the closest comparable observation-based data show a good performance overall and emphasises similarities between the models (Fig. 1). For GPP, all the models compare well to the MTE data (Jung et al., 2011) and when the directly comparable time period is used (see SI Fig. 2) the models are all within
the MTE range. The global GPP value is underlain by some regional variations between models (SI Fig. 2 and 3).

The total respiration term is similar across all the models and within a range of estimates based of the statistical upscaling of field measurements ($102 - 128$ Pg C yr$^{-1}$) (Bond-Lamberty and Thomson, 2010; Bowden et al., 1993; Luyssaert et al., 2007; Piao et al., 2010) but the partitioning between the autotrophic and heterotrophic respiration differs. Autotrophic respiration is overestimated by up to ~50% in all the models (Luyssaert et al., 2007; Piao et al., 2010), while heterotrophic respiration is
underestimated by as much as ~20% (Bond-Lamberty and Thomson, 2010). The *rh* value from Bond-Lamberty and Thomson, (2010) was reduced by 33% to account for root respiration in line with Bowden et al., (1993), and without this adjustment the discrepancy would be larger.

*is this $r_h + r_a$?*

*BNF*

[revised manuscript text omitted]

*GPP & NPP typically respond very quickly compared to ΔVeg$_C$ so this is not needed.*

*pathways*

responses  these areas that need more work. N loss is particularly challenging, as there are multiple paths (leaching, flooding, gaseous loss, fire, land use change, etc.) and forms ($N_2O$, $N_2$, etc.) of loss and each model represents these in different ways. More observational studies and syntheses of existing observations are needed to quantify the nitrogen cycle in different biomes. In particular, better constraints are needed for the N cycle response to perturbations.

All the models show a global average productivity response to increased atmospheric $CO_2$ commensurate with those
recorded in field studies. However, the regional responses and mechanisms behind this response vary widely, resulting from the interaction of the instantaneous physiological response to elevated $CO_2$ (e.g. Ainsworth and Long, (2005), which is embedded in all five models (but see Rogers et al., (2017)), with limitations imposed by temperature, water, light, and nitrogen, . This large regional variance highlights the need for a more comprehensive observational data to constrain responses to elevated $CO_2$, particularly in under-sampled regions such as
*to increased $CO_2$?*
the high arctic and tropical semi-arid regions (Song et al., 2019). Tundra and arctic responses vary widely and are associated
*are*
with the representation of BNF. In LPJ-Guess and CLM5 the responses in semi-arid tropical ecosystems is smaller than that
*Hard to follow*
*in* of temperate ecosystems and the other models, suggesting a combined effect of water- and nitrogen-limitation of soil organic matter decomposition and thus nitrogen availability that is not compensated for by changes in BNF.

The growth response to N addition across models is more varied. Two of the five models (JULES-ES and JSBACH) have
little productivity response to increased N availability, indicating that they do not have any significant limitation of the carbon cycle by N availability (Fig. 3). There are four substantial similarities between these two models (Table 1): (i) the use
*how can this be?*
of NPP to determine BNF; (ii) a direct control of NPP by N availability, whereas photosynthetic C uptake (GPP) is not directly affected by N; (iii) the use of dynamic (as oppose to prescribed) vegetation; and (iv) the assumption that N availability in pre-industrial times was sufficient to sustain the carbon cycle everywhere on land, and that observed present-
*is*
day N limitation was a result of anthropogenic changes, most notably increased $CO_2$ (Goll et al., 2017).
*is based on the assumption*
The hypothesis  of no  N limitation  prior to industrial times,

[revised manuscript text omitted]

---

## Author Response (AR2)

Reviewer 1

The comments of the reviewer are in *italics*, and author responses in blue plain type.

*I get the awkward position of being a new reviewer introduced to a paper mid-review. Like being a step-parent trying to balance my way of doing things with the fact that the kids (authors) have already developed in another system.*

We enjoyed this analogy and are doing our best to be good kids (authors) and tidy our bedroom (paper) to the satisfaction of all parents (reviewers) involved.

*Overall, this is a great paper. It is very challenging to take on a paper that not only deals with a lot of complexities and nuances within the models and observations, but also the fact that one can present the analyses/results in a gazillion different ways, making it hard for readers to absorb. The authors did an excellent job of distilling analyses, results, and interpretations, which make this paper a valuable contribution to the literature.*

Thank you for your constructive attitude, thoughtful comments, and impressive turn-around time.

*The biggest challenge is probably benchmarking N cycle impacts against a lot of C cycle measurements. Moreover, the authors do a lot of comparing model outputs to observational ranges; but, we know very well (and the authors discuss briefly in the Discussion), these magnitudes change with choice of forcing data (and other model run conditions). So, then how useful is it to make these direct comparisons? Is there not a different/better way of doing these evaluations that accounts for the fact that the end number changes so easily? The sensitivities and directions should mostly be the same no matter what forcing. I don't expect the authors to change their results at this point out of sheer exhaustion/frustration/workload related to this comment. Still, hopefully a next paper can consider this comment to advance the types of analyses done. That said, the evaluations/analyses done in this paper are much better than what is often done in other papers (e.g., let's just compare to LAI and say the difference is due to the one component that I developed in the model…).*

This has been a source of discussion for the authors from the beginning of the project when the protocol was created to discussions post-submission of the revised paper. The protocol specified CRU-NCEP for mainly pragmatic reasons, and WFDEI was a secondary forcing set to be used if groups had time. (Credit to JSBACH – this group did the second ensemble.) With hindsight, if all groups had all done the ensemble with two sets of forcing (or more) we could have had a more robust result. But as you rightly say, the reality is more simulations are not feasible now. The issue of comparability with observations is important and one which hopefully future project protocols will be able to iteratively improve.

• *Abstract*

o *Somewhere say that you ran the models offline with common spin-up and forcing protocols—this is very valuable for understanding model differences.*

Added to abstract.

o *L26-28. Maybe put something quantitative to complement the qualitative sentences, something readers can grab as take-home stats.*

The models' range of global mean % response for +N and +CO2 has been added to the abstract.

*o L29-31. It would be amazing to add why…*

We implicitly speculate in the results that it could be because if an area isn't nitrogen limited that it can respond to carbon dioxide 'fertilisation' and if it has sufficient nitrogen then more will make little or no difference. But that is speculation, so not appropriate for the abstract. This probably requires new simulations to really understand what is happening and unfortunately that is not feasible for this project.

*o L31. "better represented" is vague/unclear.*

Changed to "more accurately modelled"

*o L33. Throw away sentence. Delete.*

*o L34. "better understanding and more provision" is vague/should be more explicit.*

We have tightened and added more specifics to this sentence. And although it is a 'generic call to action' sentence, the abstract needs some sort of summary/ finishing sentence and by necessity it will be a bit vague as we are trying to summarise a study that has few clear conclusions. Therefore, we prefer to keep this sentence.

*• Introduction*

*o L41. "allowable" is that the right word? More like nothing or everything is allowed. Projections are just whatever scenarios ESMs are presented with.*

We were thinking of allowable emissions in order to meet certain targets (see Seneviratne et al. 2016 in Nature) and this was not as clear as it could have been. We have added this reference and clarified appropriately.

*o L59-61. Break up this long sentence?*

Done.

*o L60. It doesn't totally make sense why this study is limited to European centres. You commented on that in response to one of the reviewers, but it doesn't make sense in the paper. The abstract/title and everything else up to this point seems like the paper is generalizable across the global modeling community. But, then this gets inserted that throws the direction off with a jolt.*

We have removed this clarification and will leave it to the reader to wonder (or not) why these five models were used.

*o L62-63. Cite.*

Done.

*• Methods*

*o L87. Any update to Wiltshire et al forthcoming? How about a conference abstract?*

There is now a GMD paper in Discussion which we have added a reference to.

*• Results*

*o Fig 1. Cool figure.*

All credit for this figure goes to Johannes Meyerholt, who we are sure would have appreciated both the sentiment and its phrasing.

*I wonder if there is more room for artistry in it so that one can visualize the numbers and spread without having to do the math in one's head individually for each component. Could be quite powerful if you can figure it out (it's already quite powerful though, so don't get me wrong).*

We made a new version with little bar plots for each aspect, shown as a draft below. Although there are some advantages, we felt it lost the clarity that the original had.

[Figure]

- *The arrows for higher/lower than obs are nice. BUT, when you have no arrow it means either that it's within range, or that there are no obs. So, you've got some confusion there in symbology.*

We have added a note to the figure caption clarifying this.

- *Is there no uncertainty on Ndep obs?*
- *It's weird that Ndep differs between models, when they were all forced with the same amount. I guess you explain it with differences in land fraction, but it's still weird.*

It is weird, but to the best of our understanding correct and likely resulting from minor differences in the treatment of coastal grid cells, thus not especially helpful to highlight. We have changed figure 1 so it is clear(er) that N dep is prescribed.

- *What about having all the obs be a number plus/minus a number. Instead of having some be ranges. Or vice versa.*

Some of the numbers are only available as a range (rather than a +/-) so we have changed them all to a range.

- *The yellow is hard/impossible to read. Pretty much leaves me "guessing" on those numbers… (sorry, just fishing for a comment on my humor, given that you were giving out those compliments to other reviewers…).*

We have changed from yellow text to dark grey text on a yellow background, which enhances legibility. And we agree that science is more fun with puns.

- *Why no model numbers for Nmin, Nup, and Soil Ninorganic?*

We wanted to only include numbers that were available as outputs from all the models and were unequivocally comparable. We have re-considered and now included Nup, but the other two are not available for all models and thus excluded. In the process of adding Nup we had to change the method of processing, so some of the other numbers have small changes compared to the last version.

o *L164. Perhaps a slight bit more elaboration on CLM5's BNF could be useful, as it does seem to be quite different than the other 4 models.*

Added.

- *Maybe include discussion of CLM4.5 here too, given that you discussed all the models but CLM4.5?*

Added.

o *L184. Guess -> GUESS. Actually, there's inconsistency on this throughout the paper, so just do a find and replace and pick one.*

Done.

o *Section 3.2-3. Are the Song et al numbers comparable in terms of global scale, temporal scale, CO2, and climate? It seems from Song et al's Fig 1, the data are mostly geographically not where the models are being impacted most at the global scale (e.g., low for JSBACH/JULES-ES, or high for CLM's, LPJ-GUESS). If they're not comparable, then don't compare them. Throw Song et al in to the Discussion or something saying about what would be needed to make them useful.*

The Song results are specifically for 200 ppm and 50 kgN ha$^{-1}$ yr$^{-1}$ and suffer from similar issues of lack of consistent spatial distribution as any other meta-analysis does. We have added a further proviso that meta-analyses are limited by the spatial distribution of leverage points to the mean being differently distributed to those in a model. What belongs in the Results vs. the Discussion is somewhat subjective.

o *Figure 4. Maybe put somewhere on the figure that we're looking at NPP (in addition to the caption)? Would be good to have this figure stand alone.*

Done.

- *Maybe make the dots bigger? E.g., it's hard to see JSBACH and JULES-ES.*

Done.

- *Is this plus/minus latitude? Or just N. Hemisphere? If it's plus/minus, then that really isn't clear in the figure.*

We have added text to the figure legend to make it clearer that this is degrees latitude N and S (plus/minus).

*o Figure 5. The red/purple areas are hard to distinguish from one another. Same goes with the orange/yellow, though that's easier as they're more distinct geometrically.*

We have moved the order and intensity of the colours, which we hope has helped with this issue.

- *Why is there no left purple solid line?*

The purple solid line happens to be under the red solid line (as they are the same value). We have added this information to the figure caption and made the lower line dashed, to indicate they are the same value.

- *I'd consider ditching the dashed line altogether. It's really just extra information that isn't even used because the models mostly get nowhere even near the bar areas. The reader can assume the middle point.*

Good suggestion, thank you. This has 'decluttered' the plot, making it easier to read.

*o Figure 6. Cool figure. I'm confused in b and c though. They appear to be showing the N response. But, the text in L262-271 refers to the NPP response.*

Apologies that was a little unclear – the NPP refers back to figure 2f, and we've added a reference to that figure to clarify.

*o Figure 7. You introduce Fig 7d first, then 7b, and never 7a.*

We have corrected a number of small issues that have caused this: a couple of typos (7a replacing 7b), a missing reference to the figure (7d), and etc. The end result is that 7b and 7d are referenced before 7a. However, elsewhere in the figures +CO2 always comes before +N (or on the left) so although it's not optimal, we prefer to keep the figure consistent and then have the main part of the results chronological with the figure's panels.

- *Not sure if the publishing editors will pick this up, but sometimes you have a period after Fig, other times not.*

Done.

- *L280-282. I'm not following this text as it relates to the Figure. The text refers to Fig 7b. It says that JULES-ES is within range of the obs (except boreal). When I look at 7b, I don't see JULES-ES's bar inside the gray bars. Am I interpreting this incorrectly? Same goes with the statements on CLM5. You say that it's a clear outlier with a large increase in BNF. But, 7b shows a large decrease, plus it's kind of similar to LPJ-GUESS. There is an increase in 7a, but one could also just say that all the models are outliers relative to the obs, \*except\* for CLM5 in the boreal, which it actually hits.*

Our apologies, this is a typo – it makes sense if it is +CO2 (not +N) and 7a (not 7b).

- *I know CLM5 best mostly because I know FUN. So, this is a question specifically from J. Fisher to R. Fisher: how much of the CLM5 N response is due to issues with CLM's C-cycle, i.e., too much GPP/NSC/not enough Rh? It's great to see that CLM5 is going in the right directions*

*etc., but it also looks like the N cycle is hyped up on sugar, like a kid on Halloween. If you cut that GPP down, then you have less C to pay for BNF etc.*

The GP fluxes for the present day are reasonable in the CLM5, compared with relevant data products (see Lawrence et al. 2019). When fertilized with CO2 there is clearly a large increase in fluxes and also a big and rapid shift to fixation. it's not clear whether the issue here is either too much direct fertilization of PSN, or whether the issue is more with the relative cost and source switching in the CLM5 FUN parameterization. I suspect the latter. In particular, given the inflexible allocation in CLM5, one thing the plants cannot so in the case of N limitation is simply allocate more carbon to root biomass, and therefore their primary 'strategy' as the cost of direct uptake from the soil increases, is to modify fixation rates. I imagine that a more nuanced implementation of allocation patterns might well remedy this problem?

It's not clear if the Halloween metaphor can be usefully extended to include this explanation... Perhaps the enterprising kids sold their candy and invested the proceeds in their start-ups?

*o L300-302. Grammar edit.*

Rephrased.

*Good work overall! I hope my comments are useful.*

Thanks!

Reviewer 2

The comments of the reviewer are in *italics*, and author responses in blue plain type.

*I thank the authors for making their manuscript easier to follow.*

We are glad the changes we made have been effective.

*The manuscript is in much better form but I am afraid there are still some errors and seemingly incorrect misinterpretations of the data that need to be addressed before it can be accepted for publication. These are mostly minor. I also have several suggestions (some of which are personal choices on how a sentence may be phrased) that I have marked on the manuscript itself whose scanned version is attached.*

We have gone through the reviewer's in-line hand-written comments and implemented the vast majority. With up to 18 comments/edits per page on 13 pages we hope the reviewer and editor can forgive us not manually typing out each comment and responding to it individually.

*Major comments*

*1. Please report the key numbers in the abstract including that the average +CO2 response of the models is X% compared to observations (Y%), and similarly for +N response.*

We have added ranges for +N and +CO2 from the models into the abstract. However, given previous comments this reviewer made about the information necessary to include when citing observations, we are not willing to open ourselves to controversy by citing observations in the abstract where space is limited.

*2. One key analysis that is missing seems to be the comparison of late 20 th century sink. It should be pretty straightforward to compare the time series of net atmosphere-land CO2 flux from the five models with estimates from the latest Global Carbon Project (GCP) numbers for the decades of 1960s, 70s, 80s, 90s, and 2000s (https://essd.copernicus.org/articles/11/1783/2019/, their table 5). The range and average sink over the period 1960-2010, from the model, can also be reported in the abstract (since this is also a key number) and compared with the GCP's estimate.*

4/5 of the models used here are in the TRENDY paper cited above, in almost or exactly the configuration used for these simulations. Both reviewers from the first round of reviews criticized this paper for lacking in novelty. Thus we removed a 20th Century timeseries plot comparing the models to GCP data and enhanced sections with greatest novelty: the +N and +CO2 simulations. Given this change of focus, and the previous comments, we see no reason to duplicate the work of Friedlingstein et al. (2019).

*3. Right now a large fraction of the Conclusions section seems like part of the Discussion since it discusses the performance of the individual models just like in the Discussion section. I think, it would be helpful if the Conclusion section is more generalized.*

We use part of the Conclusions section to briefly summarise, in less than 130 words (~30% of the section), the key features of and differences between the models. We understand that views differ on the optimal way to present information, and respect that the reviewer's intention is to improve this paper. However, we feel this is a legitimate use of the Conclusions section as the general performance of each model is an essential conclusion of the paper.

*4. On page 9, lines 279-289. These lines discuss Figure 7a (+CO2 response) but the text (line 279) says they discuss Figure 7b (the +N response).*

Our apologies, in the many rounds of changes, from both review and co-authors, mistakes do occasionally happen. Thank you for noticing this and bringing it to our attention.

*5. Page 10, lines 300-303 read "Since the BNF in JULES is directly related to NPP, so the reduction in NUE indicates excess N in the system from mineralisation, possibly related to soil warming, in boreal regions with +CO2, leading to decreased N uptake." This sentence attempts to explain the decrease in NUE of the JULES model for the +CO2 scenario in Figure 7c. This appears to be an incorrect explanation since, I am wondering, how can the soils warm in this offline experiment which is driven with specified meteorological data, compared to the Control run. It seems there has to be some other explanation.*

The reviewer is correct that the soil warming is the same, but soil warming in the presence of elevated atmospheric carbon dioxide may respond differently to soil warming in ambient atmospheric carbon dioxide. We have rephrased this sentence to make this point clearer.

*Minor comments*

*6. The colour scheme for the five models can be better. I find it hard to differentiate between CLM 4.5 and CLM5, and CLM5 and JSBACH. Also, the yellow colour LPJ-GUESS is not readable at all on the grey boxes in Figure 1.*

We have changed the LPJ-GUESS text to grey on a yellow background to enhance legibility. We are happy to change the colour scheme in line with journal requirements if it does not already conform. However, we feel that due to the subjective nature of colour perception a change to a different colour scheme will inevitably gather some other criticism from either this reviewer or another reader.

*7. Figure 2 in SI. Do the results show model minus observations, or observations minus model?*

As is conventional, it is 'perturbation minus control'. We have clarified in the figure caption that it is model minus observations.

*8. In context of comparing observations to model results, the manuscript doesn't explain what does "upscaling" of observations means and how it is done.*

We have clarified in the text that the upscaling was done by the authors of the respective papers.

*9. The phrase "dynamic vegetation" (e.g. on line 343), I think, is meant to imply competition between different plant functional (or vegetation) types. If yes, say this explicitly since prognostic LAI, for example, is also an example of vegetation dynamics.*

We appreciate the reviewer's point that some may not know that 'dynamic vegetation' has a specific meaning in the LSM community that is distinct from 'vegetation dynamics'. We have added an explanation accordingly.

*10. Page 4, lines 118-120 read "the net ecosystem balance of N, which determines the change in the N capital available for plant growth and soil organic matter decay". This sentence doesn't read properly and, I think, is incorrect in saying that N balance is the "N capital available for plant growth". Clearly, we know that net N balance is given by [the change in the N pools] so not all of the N balance is the "capital available for plant growth".*

This sentence fragment is part of a longer sentence listing the two components important to N and terrestrial C storage. The paper comes from a N input/output perspective and this is the point of the framing in the methods. We have revised this sentence to clarify that from a model and ecosystem perspective, it is the change in the balance of input and output that determines the N capital of an ecosystem. The reviewer is correct that if looking at the change in pools, the organic N is not directly available for plant growth, however the text did not address changes in pools. A long explanation of why the input/output approach is equivalent to but subtly different from pool changes would not advance the story of the paper. Knowledgeable readers, such as the reviewer, will be familiar with the way that the N balance is generally discussed in the literature and know that this is a useful way to conceptualise changes in N, but that the detail, as with nearly all model representations, is imperfect. However, this discourse is unlikely to enhance comprehension for the average reader (say an undergraduate or PhD student), while detracting from the clarity of the paper.

*11. Equation (1) in the manuscript, and the analysis in the paper, discusses the inputs and outputs but not the changes in pool sizes. This would have been helpful in investigating how the N balance is split across the organic and inorganic pools in different models but the paper is okay without these too.*

We attach a plot of the percent change in total global C and N veg and soil pools (1996 – 2005) here for the reviewer's interest. What we can see in these plots is consistent with what is already in the paper – JSBACH and JULES-ES are similar but different to the other three. We concur that the paper is okay without this information.

[Figure]

*12. Page 11, lines 342 and 343 read "direct control of NPP by N availability, whereas photosynthetic C uptake (GPP) is not directly affected by N" in context of JULES and JSBACH showing little productivity response to increased N availability. Since NPP = GPP – Ra, I am struggling to figure, how can NPP be controlled by N availability but not GPP. The only way this can happen is Ra is controlled by N (through N content of vegetation components) in which case can this be made more clear.*

The process is more fully explained in Wiltshire et al. (2020) and Goll et al. (2017), but is not as simple as N limitation only affecting Ra. We have added these two references into the list to better

direct readers to further information about this topic. As part of a list of model similarities, we feel it is presented at an appropriate level of detail.

[revised manuscript text omitted]